# Serotonin modulates excitatory synapse maturation in the developing prefrontal cortex

Roberto Ogelman [1], Luis E. Gomez Wulschner[1], Victoria M. Hoelscher [1], In-Wook Hwang[1], Victoria N. Chang [1] & Won Chan Oh [1] ✉

Serotonin (5-HT) imbalances in the developing prefrontal cortex (PFC) are linked to long-term behavioral deficits. However, the synaptic mechanisms underlying 5-HT-mediated PFC development are unknown. We found that chemogenetic suppression and enhancement of 5-HT release in the PFC during the first two postnatal weeks decreased and increased the density and strength of excitatory spine synapses, respectively, on prefrontal layer 2/3 pyramidal neurons in mice. 5-HT release on single spines induced structural and functional long-term potentiation (LTP), requiring both 5-HT2A and 5-HT7 receptor signals, in a glutamatergic activity-independent manner. Notably, LTP-inducing 5-HT stimuli increased the long-term survival of newly formed spines ( ≥ 6 h) via 5-HT7 G$\alpha_s$ activation. Chronic treatment of mice with fluoxetine, a selective serotonin-reuptake inhibitor, during the first two weeks, but not the third week of postnatal development, increased the density and strength of excitatory synapses. The effect of fluoxetine on PFC synaptic alterations in vivo was abolished by 5-HT2A and 5-HT7 receptor antagonists. Our data describe a molecular basis of 5-HT-dependent excitatory synaptic plasticity at the level of single spines in the PFC during early postnatal development.

Brain development requires a precise interplay of neurochemicals that mediate neuronal communication and circuit formation[1,2]. 5-HT is one of the earliest detected neuromodulators[3] with cortical levels peaking within two years after birth in humans and the first postnatal week in rodents[4,5]. Changes in gestational and early postnatal 5-HT levels can arise from many causes including maternal deprivation or abuse, diets high or low in tryptophan, or the use of medications such as selective serotonin reuptake inhibitors (SSRIs) that can readily cross the placenta or be passed to offspring through breast feeding[5–8]. Disbalances of 5-HT during brain development are associated with increased risk of neurodevelopmental disorders such as autism spectrum disorder and long-lasting behavioral deficits[7,9–13], but the underlying mechanisms remain elusive.

The PFC is a brain region which plays a critical role in higher-order cognition such as social aptitude and cognitive flexibility[14]. Serotonergic (5-HTergic) axons originating from the raphe nuclei in the brainstem densely innervate the PFC and modulate its function[15]. Altered 5-HTergic signaling is thus strongly implicated in many of the PFC-dependent behavioral changes observed in neurodevelopmental disorders[7,10,12,13], and enhanced 5-HT levels in the PFC via early exposure to SSRIs are associated with behavioral impairments that last into adulthood[9,11]. While there is a clear link between 5-HTergic signaling and cognitive development, the cellular and synaptic mechanisms of 5-HT actions on neuronal plasticity and how they are altered by 5-HT imbalance in the developing PFC remain poorly understood.

Maturation and stabilization of excitatory synapses set the foundation for neural circuit formation[16–19]. Dendritic spines, the primary postsynaptic sites for excitatory synapses, form and mature to support functional circuits through activity-dependent synaptic mechanisms[20]. High levels of 5-HT in early brain development coincide with the critical period of experience-dependent excitatory synapse maturation[21]. Consistently, prior studies have implicated 5-HTergic signaling in

[1]Department of Pharmacology, University of Colorado School of Medicine, Aurora, CO 80045, USA. ✉e-mail: wonchan.oh@cuanschutz.edu

excitatory synapse development and plasticity, suggesting an essential role for 5-HT in establishing functional neural circuits[22–24]. While 5-HT acts through many distinct receptor types (5-HTRs; 7 families, 14 subtypes), 5-HT2A and 5-HT7 receptors in particular are prominent during early postnatal development, showing widespread localization on dendrites and spines in the PFC[22,25–31]. 5-HT2ARs are coupled to G$\alpha_q$ proteins and activate PKC, CaMKII, and many small molecules known to facilitate synaptic plasticity, often through concurrent Ca$^{2+}$ influx[22,26,27,29,31,32]. Stimulation of 5-HT7Rs initiates G$\alpha_s$ or G$\alpha_{12}$ dependent cascades that can activate adenylyl cyclase, PKA, and voltage-gated Ca$^{2+}$ channels, or small GTPases that are all critical for excitatory synapse development[26,27,29–31].

5-HTergic signaling is directly implicated in promoting dendritic spine plasticity[22,27,32,33], however in prior studies, synaptic plasticity was mainly induced by pharmacological activation of 5-HTRs at unidentified dendrites and/or synapses simultaneously, leaving the mechanisms underlying 5-HT-driven, synapse-specific spine plasticity unresolved. Here, we used chemogenetics, two-photon 5-HT photolysis, and pharmacological manipulations to control 5-HT signaling at both the circuit and synapse levels and examined the structure and function of dendritic spines to address the mechanisms by which 5-HT regulates individual excitatory synapses in the developing PFC.

## Results

### Serotonergic activity bidirectionally modulates excitatory synapse development in layer 2/3 pyramidal neurons of the developing PFC

We first asked whether diminished 5-HTergic transmission to the developing PFC affects excitatory synapse development on PFC layer 2/3 pyramidal neurons. To selectively inhibit 5-HTergic activity to the PFC in vivo, we injected retrograde AAV (rAAV)-DIO-hM4D(Gi) Designer Receptors Exclusively Activated by Designer Drugs (DREADDs) into the dorsal PFC of SERT-Cre mice at P1[34–36] (Fig. 1a). Using Cre-dependent retro-viral expression strategies with transgenic mice expressing Cre recombinase under the control of the SERT promoter/enhancer elements, we verified that 5-HTergic neurons originating from the dorsal raphe nucleus (DRN) form widely spread and highly dense axonal arborizations in the developing PFC in vivo (Supplementary Fig. 1a–e). We confirmed DREADD-based inhibition of 5-HTergic neuronal activity in brainstem slices containing DRN by electrophysiology following bath application of the DREADD ligand clozapine N-oxide (CNO) (Supplementary Fig. 1f–h). hM4D(Gi) or saline injected SERT-Cre animals were orally administered CNO (1 μg/g) twice daily from P6[34,36] (Fig. 1a) and apical dendrites 50–100 μm from the soma of layer 2/3 pyramidal neurons were examined at P11-15 (Fig. 1a, b). Due to a dramatic increase in spine density of PFC layer 2/3 pyramidal neurons between P11 and P15 (P11: 0.85 +/− 0.05 spines/μm, 15 cells; P15: 1.35 +/− 0.05 spines/μm, 21 cells; $p < 0.01$), all controls were age, litter, and gender-matched to DREADDs mice and experiments were performed under the same experimental conditions. We found that mice injected with hM4D(Gi)-DREADDs showed significantly lower spine density as compared to CNO only control mice at P11-15 (Fig. 1c, d). Although spine size was unaffected (Fig. 1d and Supplementary Fig. 2a, b), we observed that prefrontal spines exhibit an immature morphology with increased filopodia in hM4D(Gi)-DREADDs mice (Supplementary Fig. 3a, b). To examine functional alterations, we measured two-photon uncaging evoked excitatory postsynaptic currents (uEPSCs) from individual spines (Fig. 1b) and found that uEPSC amplitudes were smaller in hM4D(Gi)-injected mice across all spine sizes (Fig. 1e, f).

We next sought out to define how enhanced 5-HTergic transmission to the developing PFC in vivo modulates excitatory synapses by injecting rAAV-DIO-hM3D(Gq)-DREADDs into the dorsal PFC of SERT-Cre mice[34,36] (Fig. 1a). CNO bath application on hM3D(Gq)-expressing 5-HTergic neurons in the DRN resulted in decreased rheobase and depolarized resting membrane potentials (Supplementary Fig. 1i, j). In striking contrast to hM4D(Gi) experiments, spine density was significantly increased in hM3D(Gq)-injected mice compared to controls (Fig. 1g, h). We further observed an increase in large spine density (Supplementary Fig. 2c, d), with no change in spine size or morphology (Fig. 1g, h and Supplementary Fig. 3c, d). hM3D(Gq)-injected mice showed increased uEPSC amplitudes from all spines (Fig. 1i, j). Note that we targeted similar sizes of spines across groups as spine size and synaptic strength are strongly correlated[37] (Supplementary Fig. 4a, b). rAAV-DREADDs in SERT-Cre mice selectively controls 5-HT neurons in a projection-specific manner (Supplementary Fig. 5). We further verified hM3D(Gq)-mediated chemogenetic release of 5-HT in the PFC by measuring fluorescence changes of GRAB 5-HT sensor (Supplementary Fig. 6). CNO administration in vivo alone did not alter dendritic spine density or size in the PFC (Supplementary Fig. 7). DREADDs expressing pups showed normal growth rates (Supplementary Fig. 8). Together, these results demonstrate that 5-HTergic transmission in the PFC during the early postnatal period is critical for normal development of excitatory synapses on PFC layer 2/3 pyramidal neurons.

### Serotonergic stimulation induces structural long-term potentiation of individual dendritic spines on PFC layer 2/3 pyramidal neurons in early development

Our chemogenetic results established 5-HT as a crucial modulator of PFC excitatory synapse development in vivo. It remained unclear, however, whether chemogenetic manipulation of 5-HT release in the PFC affects neuronal activity of pyramidal neurons[38–40], leading to glutamatergic activity-mediated synaptic changes in the PFC (Fig. 1). We therefore measured the intrinsic membrane properties of layer 2/3 pyramidal neurons in acute PFC slices from DREADDs expressing pups (Supplementary Fig. 9a). Neither chemogenetic inhibition nor activation of PFC-projecting 5-HTergic neurons alter intrinsic properties of layer 2/3 pyramidal neurons in the PFC (Supplementary Fig. 9). These findings suggest that 5-HT may induce synaptic changes in a glutamatergic activity-independent manner. We thus examined the direct effect of 5-HT bath application on excitatory synapses and found that enhanced 5-HT signaling leads to cell-wide synaptic potentiation on both acute and organotypic PFC slices in the absence of neuronal activity (Supplementary Fig. 10)[41,42]. Consistent with the findings above, hM3D(Gq)-mediated 5-HT release in the PFC led to a similar synaptic potentiation (Supplementary Fig. 11) with no sex differences in mEPSC increases (female: 113.5 +/− 4.1%, % control, $n = 4$ cells in control, $n = 4$ cells in CNO; male: 113.0 +/− 4.5%, % control, $n = 10$ cells in control, $n = 10$ cells in CNO; female vs male: $p = 0.95$). Both female and male mice show similar changes of resting membrane potentials of DRN 5-HT neurons after CNO bath application (Supplementary Fig. 1h, j).

Next, to address the temporal dynamics and postsynaptic mechanisms of 5-HT-driven excitatory synaptic potentiation, we employed one-photon (1 P) uncaging of RuBi-5-HT (45 pulses of 1 s duration at 2 Hz)[43,44] using biolistically transfected layer 2/3 pyramidal neurons expressing GFP in organotypic PFC slices. Because no sex-related differences were observed, we used organotypic PFC slices made from either female or male pups. 5-HT was photo-released on EP11-14 PFC slices in the presence of TTX, NBQX, and CPP, while dendritic spines of layer 2/3 pyramidal neurons were monitored by two-photon imaging (Supplementary Fig. 12a) [equivalent postnatal (EP) day = postnatal day at slice culturing + days in vitro]. We found that small and medium spines undergo significant enlargement upon 1 P 5-HT uncaging (Supplementary Fig. 12b–d), indicating that 5-HT can directly induce structural long-term potentiation (sLTP) through postsynaptic signaling independently of glutamatergic activity. Yet, whether 5-HT release at individual excitatory synapses drives synapse-specific sLTP, and the molecular mechanisms underlying 5-HT-induced sLTP remained undetermined. We therefore employed two-photon

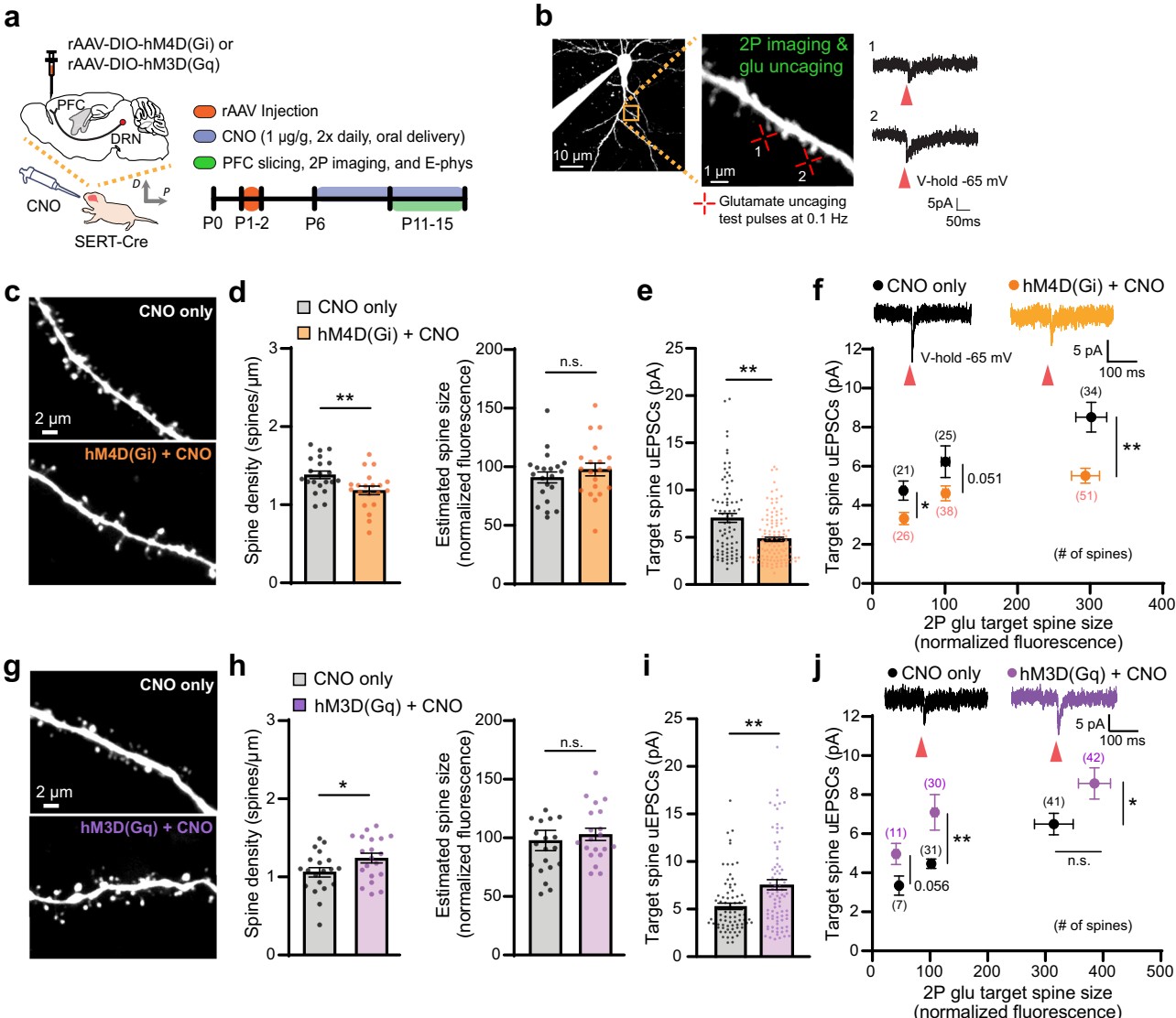

**Fig. 1 | Serotonergic activity modulates excitatory synapse development on layer 2/3 pyramidal neurons in the developing PFC. a** Schematic of retroAAV injection, oral CNO delivery, and experimental timeline. **b** 2 P images of dendrites from a PFC layer 2/3 pyramidal neuron. uEPSC traces recorded by whole-cell voltage clamp electrophysiology; a small (1) and a large spine (2) were targeted for 2 P glutamate uncaging (red arrows indicate 2 P uncaging time point). **c** Images from dendritic segments of PFC layer 2/3 pyramidal neurons from CNO only and hM4D(Gi) and CNO treated mice. Quantification of (**d**) spine density and size [CNO only: $n = 62$ dendrites, 21 cells, 4 mice (3 cells, P11-12; 18 cells, P13-15); hM4D(Gi) + CNO: 65 dendrites, 20 cells, 4 mice (3 cells, P11-12; 17 cells, P13-15); two-tailed Student's t-test (density: $p = 0.0091$; size: $p = 0.3511$)] and (**e**) uEPSC amplitudes (CNO only: $n = 80$ spines, 12 cells, 4 mice; hM4D(Gi) + CNO: 115 spines, 12 cells, 4 mice; two-tailed Mann-Whitney test $p = 0.0004$). **f** uEPSC traces from large spines. Summary of uEPSCs from small, medium, and large spines [two-tailed Mann-Whitney test (small: $p = 0.0102$; medium: $p = 0.051$; large: $p = 0.0017$)]. **g** 2 P images of dendrites from PFC layer 2/3 pyramidal neurons from CNO only and hM3D(Gq) and CNO treated mice. Quantitative analysis of (**h**), spine density and size [CNO only: $n = 72$ dendrites, 20 cells, 6 mice (6 cells, P11-12; 14 cells, P13-15); hM3D(Gq) + CNO: 69 dendrites, 20 cells, 6 mice (6 cells, P11-12; 14 cells, P13-15); two-tailed Student's t-test (density: $p = 0.047$; size: $p = 0.6088$)] and (**i**), uEPSCs (CNO only: $n = 79$ spines, 13 cells, 6 mice; hM3D(Gq) + CNO: 83 spines, 14 cells, 6 mice; two-tailed Mann-Whitney test $p = 0.0069$). **j** uEPSC traces from medium spines. Summary of uEPSCs from small, medium, and large spines [two-tailed Mann-Whitney test (small: $p = 0.0559$; medium: $p = 0.0064$; large: $p = 0.037$; large size: $p = 0.1083$)]. *$p < 0.05$, **$p < 0.01$; error bars represent SEM. n.s., not significant. Source data are provided as a Source Data file.

(2 P) 5-HT uncaging of RuBi-5-HT[45] to deliver spatiotemporally controlled patterns of 5-HT release onto individual spines, which we defined as 5-HT High-Frequency Uncaging (5-HT HFU, 30 pulses of 1 ms duration at 1 Hz, 810 nm)[42,44]. To examine the effects of 5-HT HFU on the size of stimulated target spines, we used time-lapse imaging of layer 2/3 pyramidal neurons on PFC slices at EP11-14 in the presence of TTX (Fig. 2a). Dendritic spines that received 5-HT HFU stimulation reached a significantly larger size by 15 min post 5-HT HFU and continued to exhibit sLTP throughout the imaging period compared to off-target unstimulated spines in the absence of neuronal activity (Fig. 2b–d, Supplementary Fig. 13a, and Supplementary Table 1). We

further verified that 5-HT can induce synapse-specific sLTP independently of glutamatergic signaling (Supplementary Fig. 12e, f). Spines exposed to a shifted HFU stimulus (spines > 2 μm away from uncaging point) did not undergo sLTP, indicating an input specificity of sLTP by 5-HTergic signaling (Fig. 2d). Given the age-dependent effect of 5-HT signaling onto pyramidal neurons[41,46,47], we assessed sLTP using older slices at EP15-20 and found 5-HT HFU did not trigger sLTP (Fig. 2c, d). To test whether 5-HT-dependent sLTP is pattern dependent, we employed a 0.1 Hz 5-HT Low Frequency Uncaging stimulus (5-HT LFU, 30 pulses of 1 ms duration at 0.1 Hz), which failed to enlarge stimulated spines (Fig. 2e). Interestingly, dendritic spines on layer 5 pyramidal

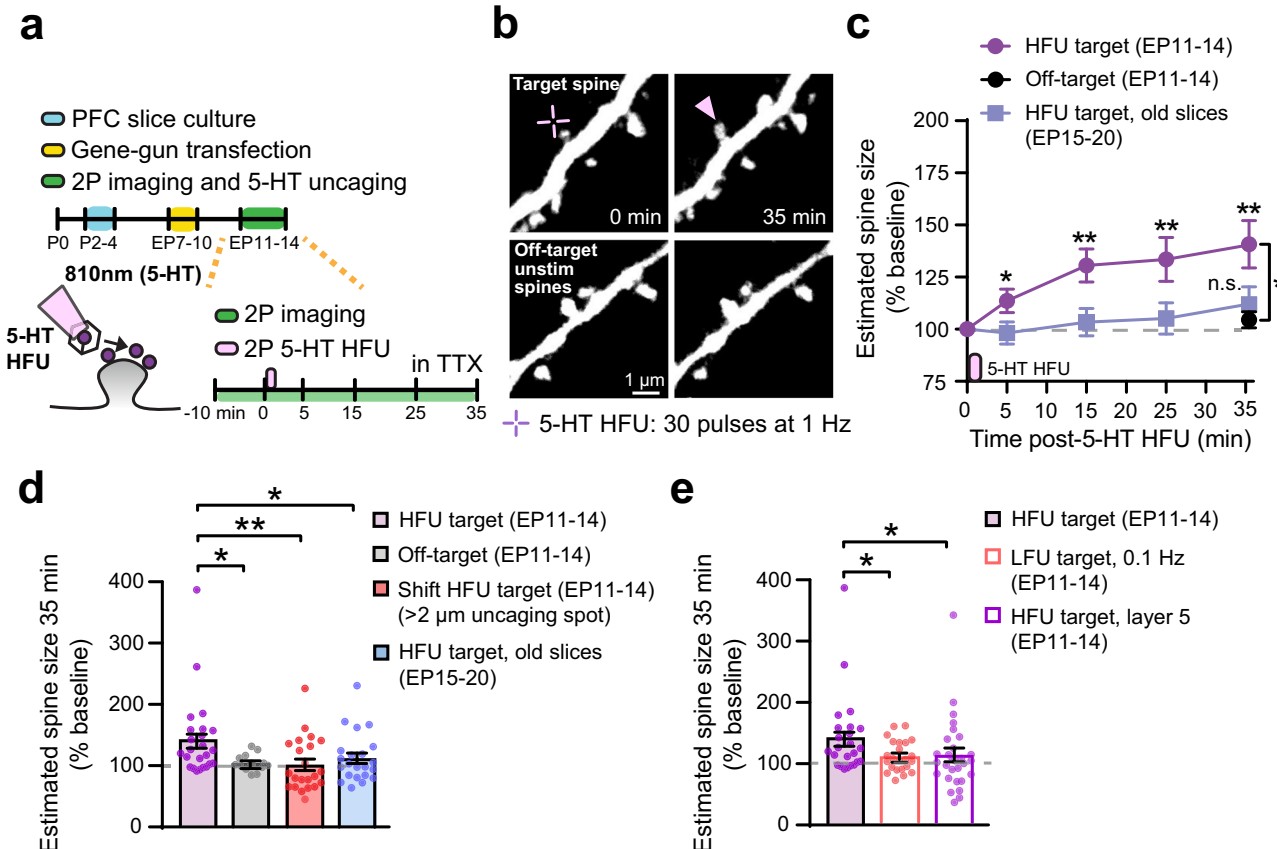

**Fig. 2 | 5-HTergic stimulation induces structural long-term potentiation of individual dendritic spines on PFC layer 2/3, but not layer 5, pyramidal neurons. a** Schematic of experimental timeline. **b** Time-lapse 2P images of a dendrite after 5-HT HFU (30 pulses of 1 ms duration at 1 Hz by 810 nm) and a control dendrite from the same neuron. Purple cross represents 5-HT HFU point; purple arrow indicates the target spine 35 min post 5-HT HFU. **c** Time course of spine size changes following 5-HT HFU (purple indicator shows 5-HT HFU time point) and (**d**) summary of spine size changes at 35 min post 5-HT HFU. [HFU EP11-14: $n = 30$ spines, 16 cells; Off-target: 13 ROIs, 13 cells; Shift HFU: 22 spines, 8 cells; HFU EP15-20: 22 spines, 12 cells; two-tailed paired t-test (HFU EP11-14, 5 min: $p = 0.0236$,

15 min: $p = 0.0006$, 25 min: $p = 0.0036$, 35 min: $p = 0.0015$; Off-target EP11-14, 35 min: $p = 0.3295$; HFU EP15-20, 35 min: $p = 0.219$); two-tailed Mann-Whitney (Off-target: $p = 0.0135$; Shift HFU: $p = 0.004$; HFU EP15-20: $p = 0.0257$)]. **e** Summary of spine size changes at 35 min post 2P 5-HT uncaging [LFU target: $n = 23$ spines, 15 cells; HFU target Layer 5: 28 spines, 14 cells; two-tailed Mann-Whitney (LFU target: $p = 0.0332$; HFU target Layer 5: $p = 0.0264$); 5-HT LFU: 30 pulses of 1 ms duration at 0.1 Hz by 810 nm]. HFU EP11-14 data (filled purple bar) are from Fig. 2d. *$p < 0.05$, **$p < 0.01$; error bars represent SEM. n.s., not significant. Source data are provided as a Source Data file.

neurons in the PFC were unaffected by 5-HT HFU (L5 HFU target: 113.8 +/− 11.4%, L5 Off-target: 104.5 +/− 5.6%; $n = 14$ cells; $p = 0.58$) (Fig. 2e). These data demonstrate that 5-HT induces sLTP independent of excitatory synaptic activity in a synapse, age, pattern, and layer-specific manner on layer 2/3 pyramidal neurons in the developing PFC.

### Postsynaptic 5-HT2A and 5-HT7 receptor signals are required for structural long-term potentiation of individual dendritic spines on layer 2/3 pyramidal neurons in the PFC

What serotonergic mechanisms underlie 5-HT-driven synaptic potentiation (5-HT sLTP) in the PFC? 5-HT2ARs express early in the PFC, localize to spines, and couple to excitatory $G\alpha_q$ proteins, which upregulate synaptic plasticity via PKC[25–28]. We tested the involvement of 5-HT2AR signaling in 5-HT sLTP and observed that blockade of either 5-HT2ARs by MDL100907 (1 μM) or PKC by Gö6983 (1 μM) completely prevented sLTP (Fig. 3a–c). Importantly, neither MDL100907 nor Gö6983 affected off-target spine size (Fig. 3b, c). 5-HT7Rs are also highly expressed in the PFC and coupled to $G\alpha_s$ and $G\alpha_{12/13}$ proteins known to be critical for synaptic plasticity[26,27,29,30]. We thus investigated the role of 5-HT7Rs and found that SB269970 (5 μM), a 5-HT7R antagonist, prevented 5-HT sLTP without affecting unstimulated spines (Fig. 3d, e). Given that activation of PKC during glutamate-

mediated LTP requires intracellular $Ca^{2+}$[48] and 5-HT7R activation leads to $Ca^{2+}$ influx[26,27], we examined whether $Ca^{2+}$ flux is necessary for 5-HT sLTP. Indeed, 5-HT HFU stimulated target spines no longer underwent sLTP in $Ca^{2+}$-free artificial cerebrospinal fluid (Fig. 3d, e). Off-target spines in 0 mM $Ca^{2+}$ showed no change in spine size (Fig. 3d, e). To further confirm the necessity for 5-HT2AR and 5-HT7R signaling in 5-HT sLTP, we used alternative antagonists and found that blockade of 5-HT2A and 5-HT7 receptors with MDL11939 (1 μM) and DR4485 (5 μM) completely abolished 5-HT-mediated sLTP without affecting off-target control spines (Fig. 3f, g). All 5-HT HFU target spines show similar initial sizes (Supplementary Table 2). Thus, we propose that extracellular $Ca^{2+}$ influx regulated by 5-HT7R signaling plays an essential role in 5-HT2AR-mediated PKC activation during 5-HT-mediated sLTP on prefrontal layer 2/3 pyramidal neurons.

### 5-HTergic stimulation induces functional long-term potentiation of individual excitatory synapses via 5-HT2A and 5-HT7 receptor signaling

Activity-dependent sLTP is positively correlated with functional strengthening[16,49,50]. To determine whether sLTP-inducing postsynaptic 5-HT signaling also leads to functional synaptic strengthening (fLTP, functional long-term potentiation), we first monitored surface

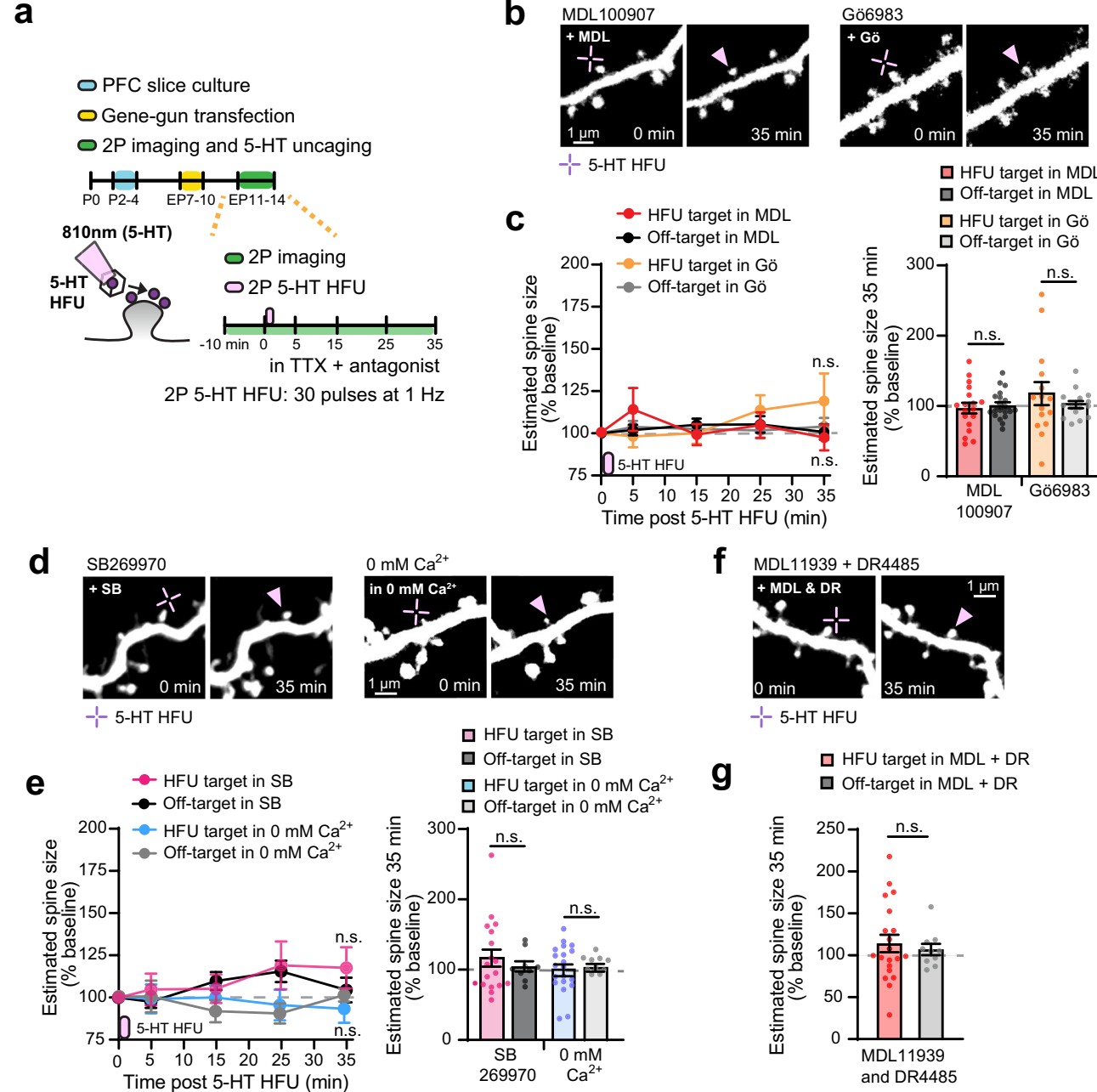

**Fig. 3 | Postsynaptic 5-HT2A and 5-HT7 receptor signals are required for structural long-term potentiation of individual dendritic spines on layer 2/3 pyramidal neurons in the PFC. a** Schematic of experimental timeline. **b** 2 P images of dendrites after 5-HT HFU in MDL100907 (1 μM) or Gö6983 (1 μM). **c** A time course plot of spine size changes following 5-HT HFU and quantitative analysis at 35 min post 5-HT HFU [HFU in MDL: $n = 18$ spines, 10 cells; Off-target in MDL: 18 ROIs, 10 cells; HFU in Gö: 15 spines, 15 cells; Off-target in Gö: 15 ROIs, 15 cells; two-tailed Student's t-test (MDL, 35 min: $p = 0.9578$; Gö, 35 min: $p = 0.387$)]. **d** 2 P images of dendrites after 5-HT HFU in SB269970 (5 μM) or 0 mM $Ca^{2+}$ ACSF. **e** A time course plot of spine size changes and quantification of spine size changes at 35 min post 5-HT HFU [HFU in SB: $n = 18$ spines, 9 cells; Off-target in SB: 9 ROIs, 9 cells; HFU in 0 mM $Ca^{2+}$: 20 spines, 10 cells; Off-target in 0 mM $Ca^{2+}$: 10 ROIs, 10 cells; two-tailed Student's t-test (SB, 35 min: $p = 0.4717$; 0 mM $Ca^{2+}$, 35 min: $p = 0.8624$)]. **f** 2 P images of dendrites after 5-HT HFU in MDL11939 (1 μM) and DR4485 hydrochloride (5 μM). **g** Spine size changes at 35 min post 5-HT HFU (HFU in MDL + DR: $n = 20$ spines, 10 cells; Off-target in MDL + DR: 10 ROIs, 10 cells; two-tailed Student's t-test MDL + DR, 35 min: $p = 0.5686$). Purple indicators in (**c, e**) show 5-HT HFU time point. Error bars represent SEM. n.s., not significant. Source data are provided as a Source Data file.

AMPARs fused with supereclictic pHluorin (SEP) following 5-HT HFU[50–52] and found that SEP-GluA2 fluorescence significantly increased at 5-HT stimulated spines that underwent sLTP (Fig. 4a–c). We next employed a combination of electrophysiology and two-color, two-photon uncaging of glutamate and 5-HT (Fig. 4a) to further confirm fLTP. After measuring baseline uEPSCs by glutamate uncaging (5-8 test pulses of 2 ms duration at 0.1 Hz, 720 nm) at target and off-target spines, a target spine was stimulated by 5-HT HFU and uEPSCs were

subsequently measured from both target and off-target spines for 30 min. 5-HT HFU at individual spines successfully induced strengthening of uEPSCs compared to off-target control spines (Fig. 4d–f). We verified that the pulsed excitation wavelength of 810 nm used for 5-HT HFU does not photo-release glutamate in our two-color, two-photon uncaging paradigm (Supplementary Fig. 14). To test whether 5-HT2AR and 5-HT7R signaling is necessary for fLTP, we repeated 5-HT HFU onto SEP-GluA2 transfected neurons in the presence of MDL100907 (1 μM)

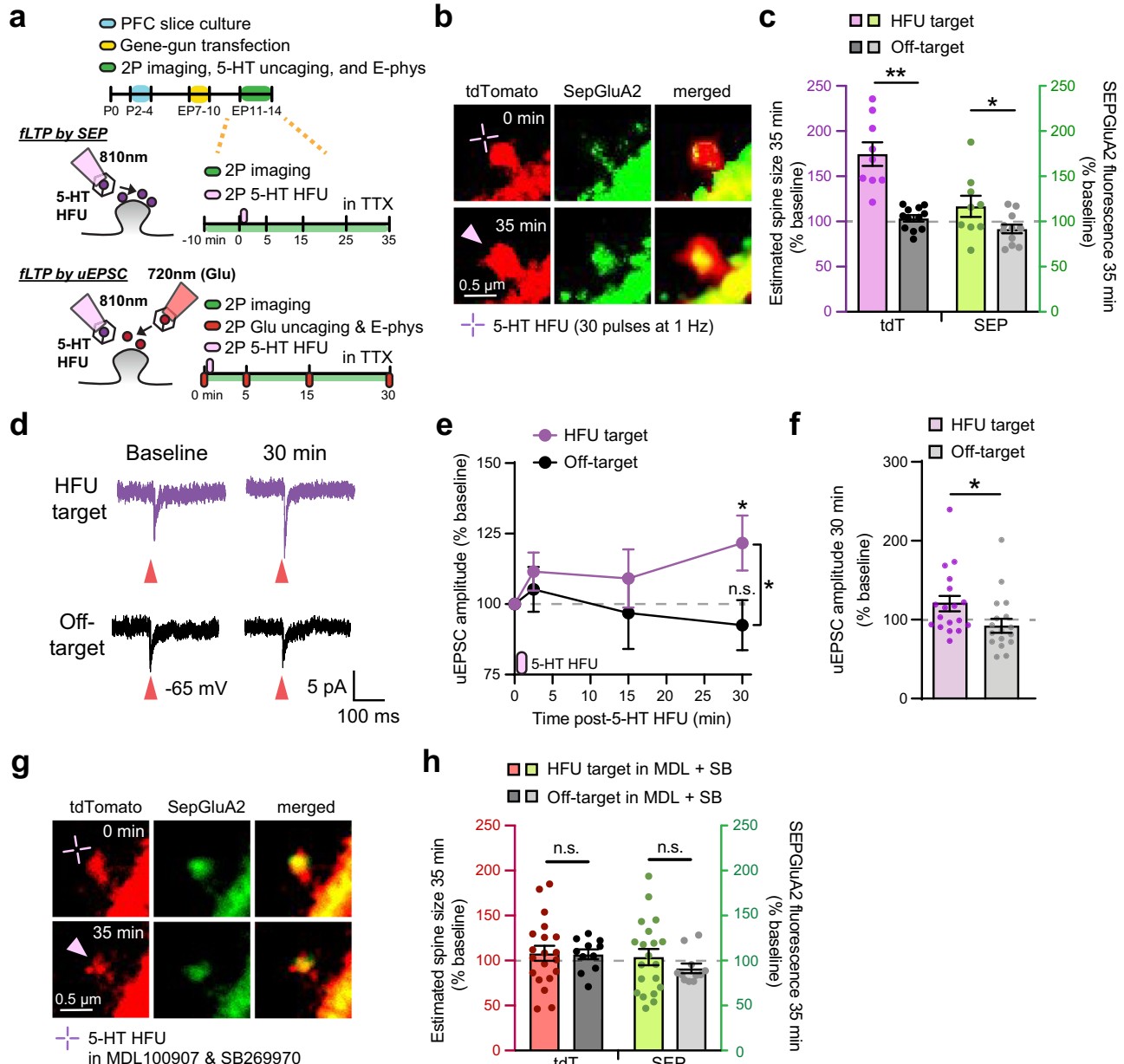

**Fig. 4 | 5-HTergic stimulation induces functional long-term potentiation of individual excitatory synapses via 5-HT2A and 5-HT7 receptors. a** Schematic of experimental design and timeline. **b** 2 P images of a dendrite from a layer 2/3 pyramidal neuron co-expressing tdTomato and SEP-GluA2 before and after 5-HT HFU. **c** Quantitative analysis of spine size and SEP-GluA2 fluorescence changes following 5-HT HFU [HFU: $n = 16$ spines, 16 cells; Off-target: 14 ROIs, 14 cells; two-tailed Student's t-test (spine size: $p < 0.0001$; SEP-GluA2: $p = 0.0414$)]. **d** uEPSC traces before and 30 min after 5-HT HFU. Red arrows indicate glutamate uncaging time point. **e** Time courses for uEPSC changes (HFU time point shown by purple indicator) and (**f**) quantification of uEPSCs at 30 min post 5-HT HFU [HFU:

$n = 18$ spines, 18 cells; Off-target: 18 spines, 18 cells; two-tailed paired t-test (5-HT HFU, 30 min: $p = 0.04$; Off-target 30 min: $p = 0.4124$); two-tailed Mann-Whitney, Off-target: $p = 0.0105$]. **g** 2 P images of a dendrite from a layer 2/3 pyramidal neuron co-expressing tdTomato and SEP-GluA2 before and after 5-HT HFU in MDL100907 (1 μM) and SB269970 (5 μM). **h** Quantitative analysis of spine size and SEP-GluA2 fluorescence changes following 5-HT HFU in MDL and SB (HFU in MDL + SB: $n = 20$ spines, 11 cells; Off-target in MDL + SB: 11 ROIs, 11 cells; two tailed Student's t-test, spine size: $p = 0.8167$; two-tailed Mann-Whitney, SEP-GluA2: $p = 0.1031$). *$p < 0.05$, **$p < 0.01$; error bars represent SEM. n.s., not significant. Source data are provided as a Source Data file.

and SB269970 (5 μM). Neither spine size nor SEP-GluA2 fluorescence increased when 5-HT2AR and 5-HT7R signaling was blocked (Fig. 4g, h). Importantly, 5-HT HFU target spines exhibit comparable initial SEP-GluA2 expression (Supplementary Table 3). Pharmacological activation of 5-HT2A and 5-HT7 receptors alone was sufficient to induce synaptic strengthening (Supplementary Fig. 15). Taken together, these results demonstrate that postsynaptic 5-HT signaling can trigger 5-HT2A and 5-HT7 receptor-dependent functional LTP (5-HT fLTP) at individual excitatory synapses in the developing PFC.

## Gαs coupled 5-HT7 receptor signaling increases the long-term stabilization of individual nascent dendritic spines on layer 2/3 pyramidal neurons in the PFC

Formation and subsequent long-term stabilization of new spines are associated with increases in spine density in early brain development[17–19]. As bidirectional manipulation of 5-HTergic activity by hM4D(Gi) and hM3D(Gq) decreased and increased the number of dendritic spines, respectively (Fig. 1), and glutamatergic LTP is known to stabilize newly formed spines[16], we hypothesized that LTP-inducing

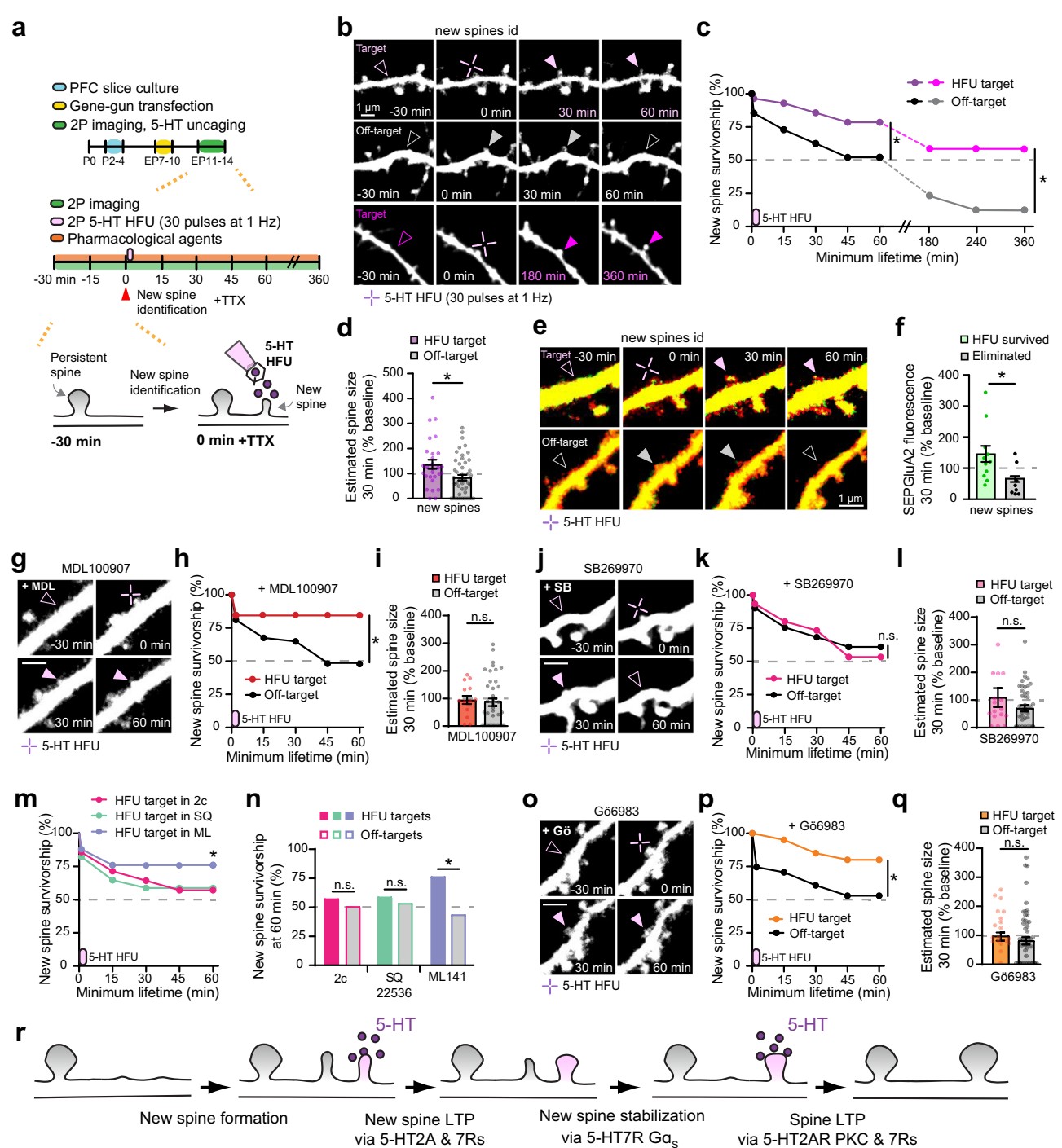

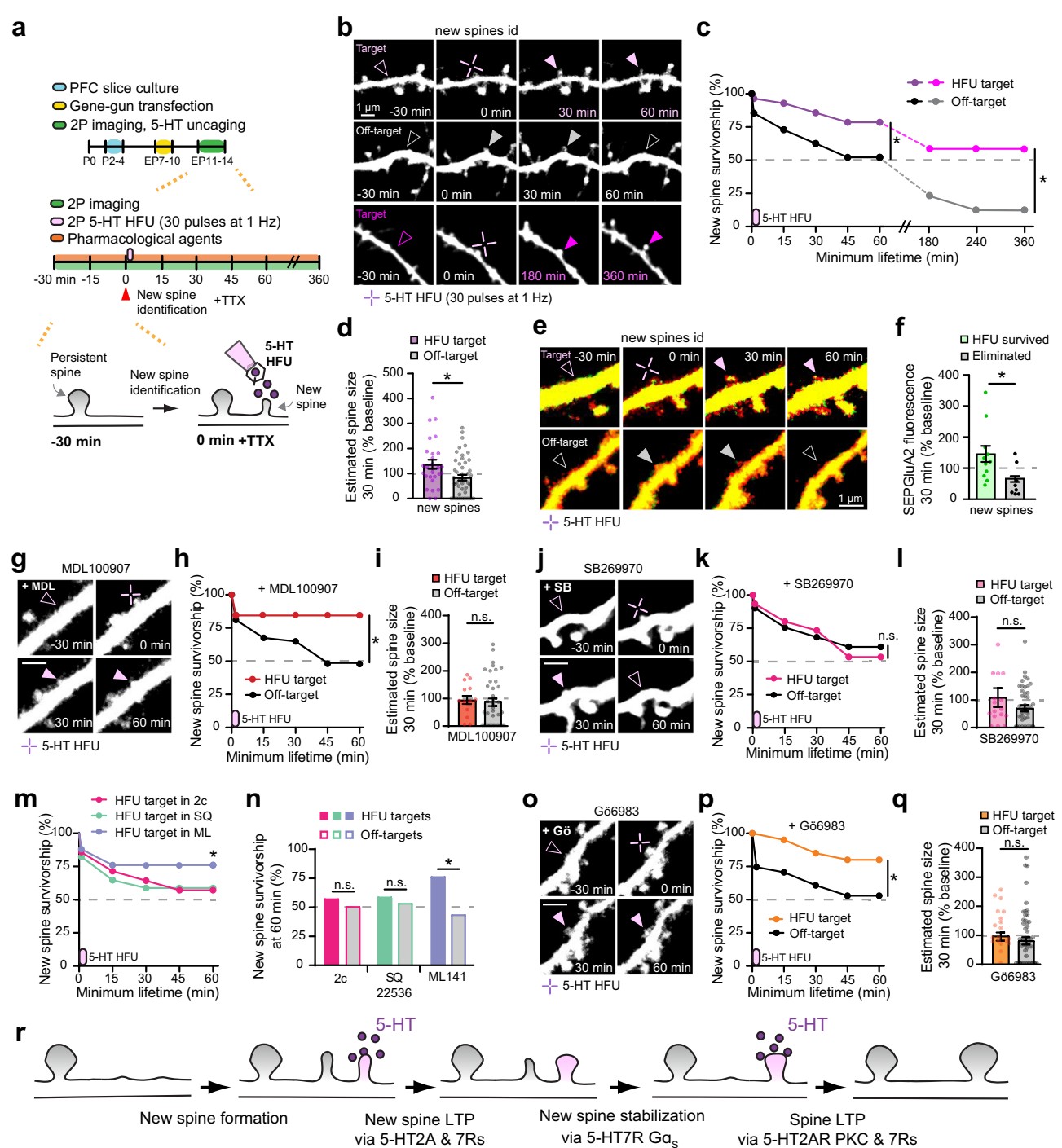

5-HT stimuli may increase the long-term stability of nascent spines, thus increasing spine density in the developing PFC. To examine this hypothesis, we carried out 5-HT HFU onto individual newly formed spines and monitored the survivorship rates of target and unstimulated off-target new spines (Fig. 5a). New spines were identified by imaging 4–6 sections of dendrites for a 30 min baseline. Once we identified nascent spines, we stimulated a target new spine by 5-HT HFU and 1-3 off target unstimulated new spines served as internal controls. TTX was added prior to 5-HT HFU to block neuronal activity. Nascent control spines in TTX and RuBi-5-HT showed a 55% survival rate over the 60 min experimental timeframe (Supplementary Fig. 13b). Notably, newly formed spines stimulated by 5-HT HFU resulted in a significantly enhanced new spine survivorship (79% survival rate) (Fig. 5b, c). Unstimulated new spines from the same neuron, serving as an internal

control, showed a similar rate of survival (52% survival rate) to control spines (Fig. 5b, c and Supplementary Fig. 13b), demonstrating a 5-HT-mediated, synapse-specific stabilization mechanism. Increases in 5-HT-dependent new spine stabilization persisted up to 360 min in the absence of neuronal activity (Fig. 5b, c). Nascent spines exposed to 5-HT HFU underwent sLTP (Fig. 5b, d), similarly to persistent spines (Fig. 2). We also monitored SEP-GluA2 fluorescence in nascent spines (Supplementary Table 3) to examine whether newly formed spines undergo fLTP. Consistent with 5-HT fLTP at persistent spines (Fig. 4), 5-HT HFU increased surface AMPAR expression of newly formed spines, notably only in survived new spines as compared to eliminated nascent spines (Fig. 5e, f). Thus, these data indicate that 5-HT LTP-inducing stimuli increase the stability of new spines in the developing PFC in a glutamatergic activity-independent manner.

**Fig. 5 | Gα$_s$ coupled 5-HT7 receptor signaling increases the long-term stabilization of individual nascent dendritic spines on layer 2/3 pyramidal neurons in the PFC. a** Schematic of experimental timeline. **b** 2 P time-lapse images of a nascent spine after 5-HT HFU (purple arrows, top) and an off-target unstimulated nascent spine (gray arrows, middle) from the same layer 2/3 pyramidal neuron over a 60 min imaging window. Bottom panels show a nascent spine that survived up to 360 min post 5-HT HFU (magenta arrows). Purple cross represents 5-HT HFU point. **c** Time courses for new spine survivorship. Purple indicator shows HFU time point [HFU up to 60 min: $n = 28$ spines, 17 cells; Off-target up to 60 min: 48 spines, 15 cells; HFU up to 360 min: 12 spines, 7 cells; Off-target up to 360 min: 17 spines, 8 cells; Fisher's exact test (60 min: $p = 0.0282$; 360 min: $p = 0.014$)]. **d** Spine size changes of new spines at 30 min post 5-HT HFU (two-tailed Mann-Whitney, $p = 0.0151$). **e** 2 P images of a nascent spine after 5-HT HFU (purple arrows) and an off-target unstimulated nascent spine (gray arrows) from the same layer 2/3 pyramidal neuron co-expressing tdTomato and SEP-GluA2. **f** SEP-GluA2 fluorescence changes of new spines at 30 min post 5-HT HFU (survived: $n = 11$ spines, 8 cells; eliminated: 8 spines, 7 cells; two-tailed Mann-Whitney, $p = 0.0404$). **g** 2 P images of a nascent spine upon 5-HT HFU in MDL100907 (1 μM). **h** Time courses for new spine survivorship (HFU in

MDL: $n = 13$ spines, 12 cells; Off-target in MDL: 37 spines, 12 cells; Fisher's exact test, $p = 0.0475$). **i** sLTP at 30 min post 5-HT HFU (two-tailed Mann-Whitney, $p = 0.6455$). **j** 2 P images of a nascent spine after 5-HT HFU in SB269970 (5 μM). **k** Time courses for new spine survivorship (HFU in SB: $n = 15$ spines, 15 cells; Off-target in SB: 41 spines, 16 cells; Fisher's exact test, $p = 0.7605$). **l** sLTP at 30 min post 5-HT HFU (two-tailed Mann-Whitney, $p = 0.6384$). **m** Time courses for new spine survivorship in 2c (15 μM), SQ22536 (100 μM), and ML141 (10 μM) (HFU in 2c: $n = 14$ spines, 8 cells; HFU in SQ: 17 spines, 12 cells; HFU in ML: 25 spines, 18 cells). **n** New spine survivorship at 60 min [Off-target in 2c: $n = 18$ spines, 7 cells; Off-target in SQ: 28 spines, 10 cells; Off-target in ML: 32 spines, 15 cells; Fisher's exact test (2c: $p = 0.7345$; SQ: $p = 1$; ML: $p = 0.0148$)]. **o** 2 P images of a new spine after 5-HT HFU in Gö6983 (1 μM). **p** Time courses for new spine stability (HFU in Gö: $n = 20$ spines, 18 cells; Off-target in Gö: 50 spines, 16 cells; Fisher's exact test, $p = 0.0349$). **q** sLTP at 30 min post 5-HT HFU (two-tailed Mann-Whitney, $p = 0.1236$). **r** A model of 5-HT-mediated stabilization and maturation of spines. Purple indicators show 5-HT HFU time point. Scale bars (**g**, **j**, **o**) = 1 μm. *$p < 0.05$, **$p < 0.01$; error bars represent SEM. n.s., not significant. Source data are provided as a Source Data file.

LTP of nascent spines following 5-HT HFU suggests a potential role of 5-HT2A and 5-HT7 receptors in enhancing new spine survivorship. We therefore tested the involvement of these two receptor subtypes. 5-HT2AR blockade by MDL100907 did not alter the new spine survival rate (85% survival rate) (Fig. 5g, h), whereas 5-HT7R blockade by SB269970 completely abolished stabilization of new spines (53% survival rate) (Fig. 5j, k). However, 5-HT sLTP of new spines was blocked by both MDL100907 and SB269970 (Fig. 5i, l), similar to results observed in persistent spines (Fig. 3). These findings suggest that structural potentiation is not a prerequisite for 5-HT-mediated new spine stabilization. 5-HT7Rs are coupled to Gα$_s$ and Gα$_{12/13}$ proteins[26,27,31]. To distinguish between these two pathways under 5-HT7R-dependent new spine stabilization, we took advantage of compound 2c, a selective biased antagonist for 5-HT7R Gα$_s$ that does not inhibit Gα$_{12/13}$ signaling[53], and found inhibiting the Gα$_s$ pathway with 2c (15 μM) blocked new spine stabilization (57% survival rate) (Fig. 5m, n). Blockade of adenylyl cyclase, a downstream enzyme of Gα$_s$ signaling, by SQ22536 (100 μM) also prevented new spine survivorship (59% survival rate) (Fig. 5m, n), confirming the involvement of Gα$_s$ signaling in new spine stabilization. Inhibiting Cdc42, a small GTPase activated by Gα$_{12/13}$ signaling, with ML141 (10 μM), did not affect 5-HT induced new spine survivorship (76% survival rate) (Fig. 5m, n). In line with downstream 5-HT2AR signaling pathways, PKC inhibition with Gö6983 abolished 5-HT sLTP of new spines without affecting new spine survivorship (80% survival rate) (Fig. 5o, p). None of these drugs altered the stabilization rate of internal control new spines (Fig. 5n, p). These results strongly suggest a 5-HTR subtype-specific mechanism in which Gα$_s$-dependent adenylyl cyclase activation by 5-HT7Rs, but not 5-HT2ARs, stabilizes nascent spines (Fig. 5r) in the developing PFC.

**Fluoxetine increases PFC layer 2/3 excitatory synapse strength and density via 5-HT2A and 5-HT7 receptor activation**
Fluoxetine (FLX) is an antidepressant that blocks presynaptic reuptake of 5-HT, consequently increasing 5-HT concentrations in the synaptic cleft[54]. Early FLX exposure is linked to behavioral deficits and neurodevelopmental disorders[7,10,12], yet associated alterations in excitatory synapse development in the PFC remain unexplored. Based on our in vivo chemogenetic manipulations resulting in increased spine density and strength by hM3D(Gq), and 5-HT HFU experiments demonstrating 5-HT-dependent LTP and stabilization, we hypothesized that increased 5-HT signaling after FLX administration would result in similar synaptic density and strength enhancement. To examine how PFC excitatory synapses are affected by enhanced 5-HT levels via FLX in vivo, we orally administered FLX (10 μg/g) to pups starting at P2 and assessed excitatory synapses of layer 2/3 pyramidal neurons at P11-15 (Fig. 6a). Notably, we observed a significant increase

in spine density from FLX-treated pups compared to saline-treated, age and gender-matched controls, without affecting spine size (Fig. 6b, c and Supplementary Fig. 16a, b). Large spines from FLX-treated mice showed significantly larger uEPSCs (Fig. 6d, e and Supplementary Fig. 4c). FLX (10 μg/g) treatment to older pups aged P16-24 did not induce any alterations in excitatory synapse structure and function (Supplementary Fig. 17). Together with the age-dependent 5-HT sLTP findings from PFC slices in Fig. 2c, d, these results suggest an age-specific 5-HTergic effect on excitatory synapses.

Given the role of 5-HT2AR and 5-HT7R signals in PFC synapse development, we next assessed whether excitatory synaptic alterations following FLX exposure from P2-15 are abolished by inhibiting 5-HT2AR and 5-HT7R signaling. Mice were orally administered FLX and antagonists for 5-HT2ARs (MDL11939, 0.3–0.5 μg/g daily) and 5-HT7Rs (DR4485 hydrochloride, 0.3–0.5 μg/g daily) (antagonists were administered 10–20 min prior to FLX; Fig. 3f, g, Fig. 6a)[55,56]. We found that increased spine density observed in FLX-treated mice was restored in FLX and antagonist-treated mice to levels comparable to spines in vehicle-treated control mice (Fig. 6f, g and Supplementary Fig. 16c, d). Blockade of 5-HT2AR and 5-HT7R in combination with FLX decreased uEPSCs of large spines compared to those in control mice (Fig. 6h, i and Supplementary Fig. 4d), whereas 5-HT2AR and 5-HT7R antagonists alone had no effect on spine density, size, or uEPSCs (Fig. 6j–m and Supplementary Figs. 4e and 16e, f). All drug and vehicle treated animals were handled equally and FLX and antagonist-treated pups showed normal body weight gain (Supplementary Fig. 18). Together, these results demonstrate that in agreement with our slice work, FLX-mediated excitatory synapse alteration requires 5-HT2AR and 5-HT7R signaling in the developing PFC in vivo.

## Discussion
5-HTergic transmission is a critical mediator of PFC development, evidenced by a growing body of literature highlighting the long-term behavioral consequences of 5-HT imbalances during gestation and early postnatal life[7,9–13]. While it is widely accepted that 5-HT acts as a key neuromodulator regulating neuronal connectivity in the cortex[10,26,30,31,42], the underlying synaptic mechanisms remain unclear. We demonstrate a synapse-specific role for 5-HTergic signaling in excitatory synapse maturation during the critical period of circuit formation in early PFC development. Our chemogenetic and pharmacological manipulations of 5-HTergic signaling in vivo show a critical function for 5-HT during early postnatal development in shaping dendritic spine maturation and function. Furthermore, our study establishes a spatiotemporally controlled method that induces 5-HT-dependent and synapse-specific strengthening and stabilization of excitatory synapses on PFC layer 2/3 pyramidal neurons in a

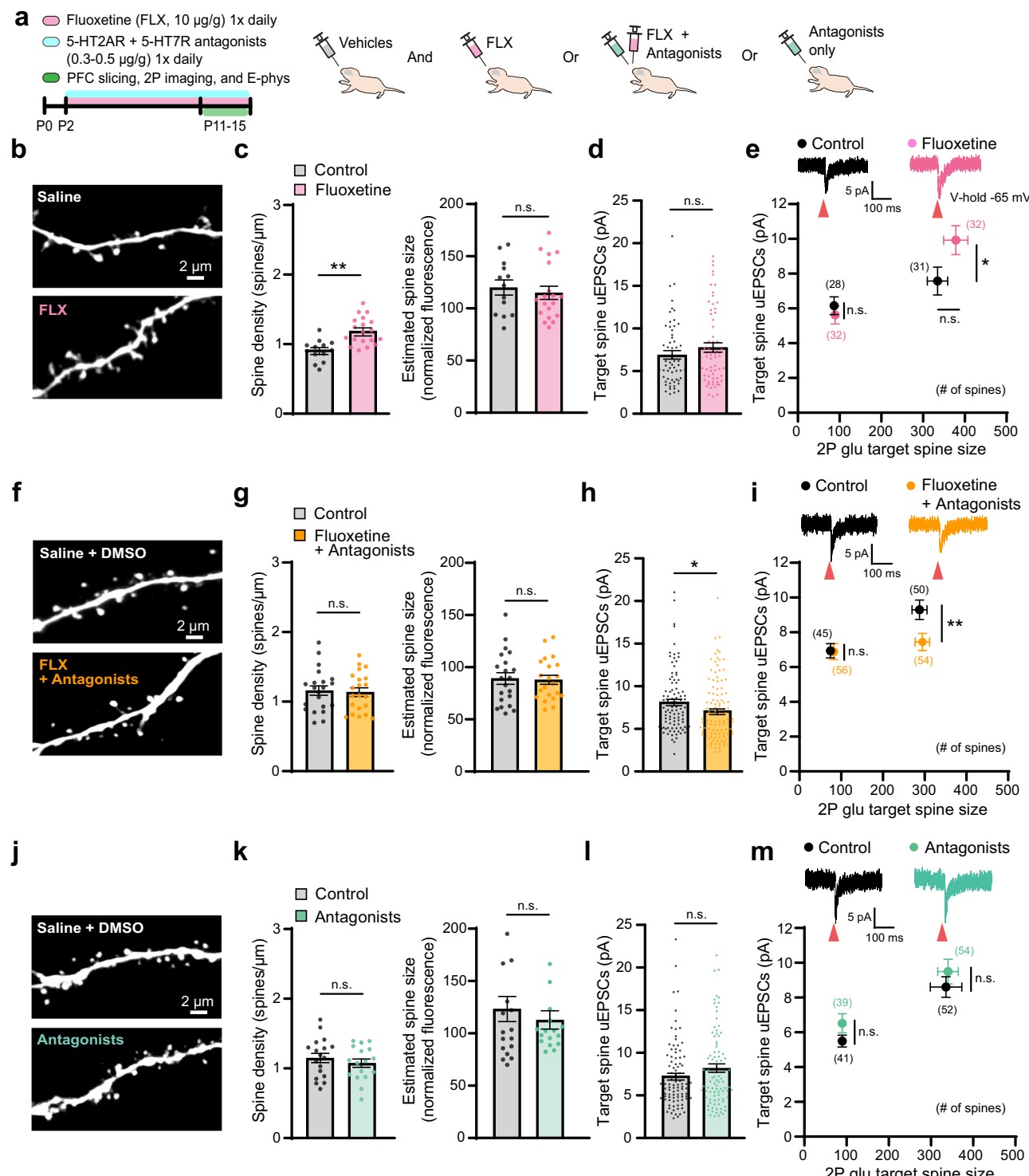

glutamatergic-activity independent manner. This two-photon 5-HT uncaging method allowed us to further define the age, pattern, and layer-specificity of 5-HT2A and 5-HT7 receptor-mediated dendritic spine plasticity in the developing PFC.

Notably, our two-photon uncaging manipulations suggest the synaptic mechanisms underlying 5-HT-driven spine maturation in early development are glutamatergic activity-independent. While previous studies have shown a critical role of 5-HT on cell-wide neuronal plasticity[22,27,32,33], our results demonstrate that 5-HT can directly induce excitatory synapse plasticity at individual excitatory synapses in the absence of glutamatergic signaling. Our chemogenetic manipulations

did not affect intrinsic properties of layer 2/3 PFC pyramidal neurons, further suggesting that our in vivo DREADDs results could be due to a direct effect of 5-HT on synapse maturation rather than activity-dependent changes on excitatory synapses by altered excitability of pyramidal neurons. Although these results support 5-HT-driven plasticity, we cannot rule out the possibility that altered density or strength of dendritic spines may be affecting pyramidal cell firing[57], in turn inducing glutamatergic activity-dependent synaptic plasticity. Furthermore, studies have shown that neurons projecting out of the DRN may co-release 5-HT and glutamate during punishing or rewarding stimuli[58,59]. It is possible that 5-HT-driven synaptic maturation in vivo is

**Fig. 6 | Fluoxetine-mediated alteration of PFC layer 2/3 excitatory synapse development requires 5-HT2A and 5-HT7 receptor activation. a** Schematic of experimental timeline and oral delivery conditions. **b** 2 P images of dendritic segments of PFC layer 2/3 pyramidal neurons from saline or FLX treated mice. **c** Summary of spine density and size [Control: $n = 50$ dendrites, 13 cells, 5 mice (9 cells, P11-12; 4 cells, P13-15); FLX: 69 dendrites, 19 cells, 5 mice (12 cells, P11-12; 7 cells, P13-15); two-tailed Student's t-test (density: $p = 0.0002$; size: $p = 0.6054$)]. **d** Quantitative analysis of uEPSCs (Control: $n = 59$ spines, 9 cells, 3 mice; FLX: 64 spines, 11 cells, 3 mice; two-tailed Mann-Whitney test $p = 0.3789$). **e** uEPSC traces from large spines and summary of uEPSCs from small-medium and large spines [two-tailed Mann-Whitney test (small-medium uEPSCs: $p = 0.3172$; large uEPSCs: $p = 0.0394$; large size: $p = 0.2757$)]. **f** 2 P images from dendritic segments of PFC layer 2/3 pyramidal neurons from saline and DMSO or FLX, MDL11939, and DR4485 treated mice. **g** Summary of spine density and size [Control: $n = 64$ dendrites, 21 cells, 4 mice (11 cells, P11-12; 10 cells, P13-15); FLX + Antagonists: 67 dendrites, 21 cells, 4 mice (11 cells, P11-12; 10 cells, P13-15); two-tailed Student's t-test (density: $p = 0.8025$; size: $p = 0.8574$)]. **h** Quantitative analysis of uEPSCs (Control: $n = 95$ spines, 15 cells, 4 mice; FLX + Antagonists: 110 spines, 17 cells, 4 mice; two-tailed Mann-Whitney test $p = 0.0151$). **i** uEPSC traces from large spines and quantitative analysis of uEPSCs from small-medium and large spines [two-tailed Mann-Whitney test (small-medium: $p = 0.4354$; large: $p = 0.005$)]. **j** 2 P images from dendritic segments of PFC layer 2/3 pyramidal neurons from saline and DMSO or MDL11939 and DR4485 treated mice. **k** Summary of spine density and size [Control: $n = 60$ dendrites, 18 cells, 4 mice (9 cells, P11-12; 9 cells, P13-15); Antagonists: 60 dendrites, 17 cells, 4 mice (9 cells, P11-12; 8 cells, P13-15); two-tailed Student's t-test (density: $p = 0.2945$; size: $p = 0.4905$)]. **l** Quantitative analysis of uEPSCs (Control: $n = 93$ spines, 15 cells, 4 mice; Antagonists: 93 spines, 12 cells, 4 mice; two-tailed Mann-Whitney test $p = 0.2801$). **m** uEPSC traces from medium spines and quantitative analysis of uEPSCs from small-medium and large spines [two-tailed Mann-Whitney test (small-medium: $p = 0.2846$; large: $p = 0.4237$)]. Red arrows in (**e**, **i**, **m**) indicate glutamate uncaging time point. *$p < 0.05$, **$p < 0.01$; error bars represent SEM. n.s., not significant. Source data are provided as a Source Data file.

mediated by a combination of direct 5-HTergic signaling onto dendritic spines and in turn, indirect alterations of glutamatergic signaling that further affect synaptic properties. Future experiments will examine glutamatergic release after chemogenetic manipulations of 5-HT neurons to identify the contribution of glutamate to prefrontal synaptic plasticity in vivo.

5-HT2A and 5-HT7 receptors are distributed throughout the PFC in early postnatal development and have been shown to localize postsynaptically on dendrites and spines[22,25–31]. We observed 5-HTergic activation at pre-existing spines induces structural and functional LTP through both postsynaptic 5-HT2A and 5-HT7 receptor signaling. Similarly, newly formed spines undergo 5-HT-mediated spine enlargement that requires 5-HT2A and 5-HT7 receptors as they subsequently stabilize. Intriguingly, however, nascent spines require only 5-HT7 receptor signaling through $G\alpha_s$ to increase their long-term stability upon 5-HTergic activation, in that new spines exhibit enhanced survivorship without structural potentiation in the absence of 5-HT2A receptor signaling. These results indicate that, unlike glutamatergic activity-dependent new spine stabilization primarily by NMDA receptor activation[16], synaptic potentiation is not a pre-requisite for 5-HT-driven new spine stabilization. Determining if new spines undergo LTP through the same downstream 5-HTergic mechanisms observed for persistent spines will be interesting to investigate in the future[10,19,26,30,31].

Our work directly interrogating 5-HT receptor subtype-specific synaptic mechanisms supports a model in which newly formed spines stabilize through $G\alpha_s$-mediated mechanisms upon 5-HT7 receptor activation. As these surviving spines mature, signaling from both 5-HT7 and 5-HT2A receptors drives synapse maturation via PKC activation[48]. What might explain the differences in 5-HT receptors required for nascent spine stabilization versus pre-existing spine LTP? In small, newly formed spines, increased $Ca^{2+}$ entry through 5-HT7 receptor-dependent, Adenylyl Cyclase-mediated voltage-gated $Ca^{2+}$ channel (VGCC) activation may be sufficient to drive new spine stabilization[60–62]. In mature spines, additional PKC activation via 5-HT2A receptor signaling, which may require the increased $Ca^{2+}$ entry from 5-HT7 receptors, induces structural enlargement and functional strengthening of prefrontal spines in layer 2/3 pyramidal neurons. In support of this notion, 5-HT-driven spine potentiation is abolished in $Ca^{2+}$-free artificial cerebrospinal fluid, confirming that extracellular $Ca^{2+}$ entry is necessary for 5-HT-dependent synaptic strengthening. Future work will further delineate the distinct downstream mechanisms involved in 5-HT-dependent spine stabilization and LTP[61–65].

The age-dependent effects observed in our two-photon 5-HT uncaging results align with prior studies establishing a switch in 5-HT receptor signaling in postnatal rodent development. During the first two postnatal weeks, 5-HTergic signaling onto PFC pyramidal neurons induces 5-HT2A and 5-HT7 receptor-dependent depolarization[41,46,47].

Contrarily, 5-HT begins to hyperpolarize PFC neurons via $G\alpha_i$ coupled inhibitory 5-HT1A receptor mechanisms after the second postnatal week[41]. Similarly, our results indicate that fluoxetine (FLX) administration to older pups aged P16-24 does not induce the same alterations in excitatory synapse properties as FLX administered from P2-15, confirming an age-dependent effect of 5-HTergic signaling on prefrontal synapse development. These findings further indicate there may be decreased 5-HT2A and 5-HT7 receptor signaling after the second postnatal week. Additionally, we observed a depression of dendritic spine strength when 5-HT2A and 5-HT7 receptors were blocked from P2-15 during concurrent FLX administration, suggesting there may be an increased role of inhibitory 5-HT receptors such as 5-HT1A or 5-HT1B on synaptic weakening in the absence of excitatory 5-HTergic receptor signaling[26,27,31,41,66,67]. Similarly, the lack of structural potentiation observed in layer 5 pyramidal neurons could be explained by a lower ratio of excitatory 5-HT2A or 5-HT7 receptor to inhibitory 5-HT1A/B receptor expression compared to the ratio of these receptors in layer 2/3. The role that other 5-HTergic receptors, such as other 5-HT2 receptor isoforms, 5-HT1B, 5-HT3, and 5-HT4 receptors (among the many other classes) play in excitatory plasticity in the developing PFC is still unclear[26,27,31]. The classes of 5-HT receptors involved in plasticity in other brain regions, on apical versus basal dendrites, or during different developmental stages will be interesting avenues of research for future studies. Furthermore, 5-HTergic receptor signaling and expression is known to differ by gender and studies have shown that 5-HT can have opposing behavioral effects on mature female versus male mice, particularly in models of anxiety or depression[68–72]. While we did not find any sex differences in 5-HT neuronal activity by chemogenetic manipulations at this early developmental stage, these data can be used to determine the potential sex effects on long-term behavioral and synaptic deficits as these mice mature. Whether sex differences in 5-HTergic signaling are more pronounced as sexual maturity begins will be an exciting future avenue of investigation.

As our 5-HT uncaging experiments demonstrate that 5-HTergic signaling can enlarge spine size, we were intrigued by our in vivo experimental results showing enhanced excitatory synapse strength yet a lack of spine size changes. The absence of structural alteration may be explained by one or more of the following possibilities. First, the long-term 5-HTergic manipulations may have different effects on synapses than that of short-term stimulation onto spines. Second, our in vivo approach cannot be confined to excitatory postsynapses, and thus we lack the ability to target postsynaptic 5-HT receptors in PFC layer 2/3 pyramidal neurons. Third, while our uncaging experiments elucidate glutamatergic-independent mechanisms of 5-HT-mediated plasticity on PFC slices, our work does not exclude a possible contribution of glutamatergic signaling after 5-HTergic neuron manipulations in vivo. Potential co-release of glutamate or varied levels of glutamatergic signaling throughout the PFC could explain the

functional excitatory changes observed. The molecular mechanisms underlying altered spine strength of similar-sized spines in DREADDs and FLX-mediated manipulations will be an exciting future direction for these studies. Specifically, how are glutamatergic AMPA receptor properties changed, and is subunit composition altered within dendritic spines after 5-HT stimulation[73,74]? Are AMPARs packing more densely within spines, and can 5-HTergic signaling change nanodomain number or size within the postsynaptic density[75,76]?

We also found it interesting that oral administration of 5-HT2A and 5-HT7 receptor antagonists alone had no effect on spine structure or function. Our working model predicts a decrease in spine density and/or strength by inhibiting 5-HT2A and 5-HT7 receptor signaling. One explanation for this seeming discrepancy is that oral delivery blocks 5-HT2A and 5-HT7 receptors throughout the entire nervous system, including directly onto 5-HTergic neurons that project to the PFC, and within many brain regions that innervate the PFC[26,28,31,55,56]. Circuit level signaling changes from diminished 5-HT2A and 5-HT7 receptor activation may differ from local inhibition at a synapse level within the PFC. Additionally, antagonists were delivered once daily, which may not have been sufficient to induce excitatory synapse alterations. Prolonged blocking of 5-HT2A and 5-HT7 receptor signaling specifically in layer 2/3 pyramidal neurons in the PFC would likely result in a decrease in spine density and strength as our model predicts.

Taken together, we clarify the cellular and molecular basis of 5-HT-dependent plasticity at the level of single dendritic spines and highlight the significance of 5-HT neuromodulation in shaping PFC circuitry during early brain development. SSRIs such as FLX are commonly taken throughout the world by pregnant people, affecting 5-HTergic signaling in offspring[7,8]. Infants born following perinatal exposure to SSRIs have a higher risk for neurodevelopmental disorders[7,8], however, the underlying mechanisms remain unknown. Our findings provide experimental evidence that postnatally FLX-treated animals exhibit deficits in synapse maturation in the PFC via 5-HTergic activities resulting from enhanced 5-HT2A and 5-HT7 receptor signaling. Understanding the mechanisms behind 5-HTergic plasticity and signaling during this critical time-period will help develop therapeutics that will more selectively target 5-HT receptors in specific brain regions or neuronal subpopulations. Furthermore, our findings may provide new avenues in the pharmacological treatment of patients who have been exposed to antidepressants during the perinatal period.

## Methods

### Preparation of acute brain slices
Mice were acquired from Jackson Laboratories (C57BL/6NJ, wild type, and B6.129(Cg)-Slc6a4tm1(cre)Xz/J, SERT-Cre). All the experiments involving acute coronal slices of the prefrontal cortex (PFC) and the brainstem that contains dorsal raphe nucleus (DRN) were prepared from both male and female wild type and SERT-Cre mice at P6-24 in accordance with the Institutional Animal Care and Use Committees of the University of Colorado on Anschutz Medical Campus and National Institutes of Health guidelines. Mice were anesthetized with isoflurane and sacrificed by decapitation. The brain was removed from the skull and rapidly placed in ice-cold cutting solution containing (in mM): 215 sucrose, 20 glucose, 26 NaHCO$_3$, 4 MgCl$_2$, 4 MgSO$_4$, 1.6 NaH$_2$PO$_4$, 1 CaCl$_2$ and 2.5 KCl. Cortical slices (300 μm thick) were prepared using a VT1000S (Leica) vibrating microtome. Slices were incubated at 32 °C for 30 minutes in a holding chamber containing 50% cutting solution and 50% artificial cerebrospinal fluid (ACSF) containing (in mM): 127 NaCl, 25 NaHCO$_3$, 25 D-glucose, 2.5 KCl, 1.25 NaH$_2$PO$_4$, 2 CaCl$_2$, and 1 MgCl$_2$. After 30 min, this solution was replaced with ACSF at room temperature. Slices were allowed to recover for at least 1 h in ACSF before imaging and/or recording. For two-photon imaging and electrophysiology experiments, slices were transferred to a submersion-

type, temperature-controlled recording chamber (TC-324C, Warner Instruments) and perfused with ACSF. Imaging and recordings were performed at 30 °C. All solutions were equilibrated for at least 30 min with 95% O$_2$ / 5% CO$_2$.

### Preparation and transfection of organotypic PFC slice cultures
Organotypic slice cultures from the PFC (400 μm thick) were prepared from postnatal day 2 (P2)-P4 male and female pups as described previously[77], in accordance with the Institutional Animal Care and Use Committees of the University of Colorado on Anschutz Medical Campus and National Institutes of Health guidelines. Mice were acquired from Jackson Laboratories (wild type and SERT-Cre). Slices were transfected with tdTomato, EGFP, and/or SEP-GluA2 2–7 days prior to two-photon imaging and electrophysiology using biolistic gene transfer (180 psi)[50,78]. A total of 10–15 μg of tdTomato, 10 μg of EGFP, and/or 15 μg of SEP-GluA2 were coated onto 7–9 mg of 1.6 μm diameter gold particles. The age of slice culture is reported as equivalent postnatal (EP) day (i.e., postnatal day at slice culturing + days in vitro).

### Two-photon imaging
Imaging was performed on layer 2/3 or layer 5 pyramidal neurons at depths of 20–50 μm of dorsolateral PFC (PFC slice cultures at EP11-20 or PFC acute slices at P11-24) using a two-photon microscope (Bruker, Olympus LUMPLFLN60XW) with a pulsed Ti:sapphire laser (MaiTai HP, Spectra Physics) tuned to 920 nm (3–5 mW at the sample) or with a HighQ-2 laser (Spectra Physics) tuned to 1045 nm (3–5 mW the sample). The microscope and data acquisition were controlled with Prairie View (Bruker) and 302RM (Conoptics). For time-lapse imaging, neurons were imaged at 5–15 min intervals at 30 °C in recirculating ACSF aerated with 95% O$_2$/5% CO$_2$ (~310 mOsm, pH 7.2) with 2 mM CaCl$_2$, 1 mM MgCl$_2$, 0.001 mM tetrodotoxin (TTX), 2.5 mM 4-methoxy-7-nitroindolinyl-caged-L-glutamate (MNI-Glutamate, Tocris) and/or 0.1 mM (bis (2,2′-Bipyridine-N,N′) trimethylphosphine)-5-hydroxytryptamine ruthenium$^{2+}$) dihexafluorophosphate complex (RuBi-5-HT, Tocris or Abcam). For each neuron, image stacks (512 × 512 pixels, 0.048 μm/pixel) with 1 μm z-steps were collected from one to six segments of secondary and/or tertiary apical dendrites 50–100 μm from the cell body of layer 2/3 or 5 pyramidal neurons. GRAB 5-HT1.0 sensor imaging was performed on layer 2/3 pyramidal neurons of acute PFC slices at P13-15 using a 920 nm tuned Ti:sapphire laser (MaiTai HP, Spectra Physics) in ACSF containing (in mM): 2 Ca$^{2+}$ and 1 Mg$^{2+}$[79]. All images shown are maximum projections of 3D image stacks after applying a median filter (2×2) to the raw image data.

### One-photon and two-photon uncaging
One-photon RuBi-5-HT uncaging experiments were performed using 45 pulses of full-field illumination (470 nm, 7–8 mW at the sample, Thorlabs) of 1 s duration at 2 Hz to release 5-HT over the entire slice preparation. High-frequency two-photon uncaging of RuBi-5-HT (5-HT HFU) was performed using 30 pulses (810 nm, 5-8 mW at the sample) of 1 ms duration at 1 Hz. Low frequency two-photon uncaging of RuBi-5-HT (5-HT LFU) was performed using 30 pulses (810 nm, 5–8 mW at the sample) of 1 ms duration at 0.1 Hz. For two-photon RuBi-5-HT stimulation, no more than one spine on two separate dendritic regions of interest was stimulated per cell, at least 50 μm apart. One to five regions of interest (ROIs, > 50 μm distant from a target spine) were used as an off-target internal control to monitor spine size changes, new spine stabilization, or excitatory synapse strength without nearby two-photon 5-HT stimulation. Two-photon uncaging of MNI-Glutamate was employed to assess the strength of excitatory synapses, 5–8 individual laser pulses (720 nm, 7–10 mW at the sample) of 2 ms duration were delivered at 0.1 Hz[78]. To measure the strength of spines over time after 5-HT HFU, two-color, two-photon uncaging of MNI-Glutamate and RuBi-5-HT was performed by assessing excitatory

synapse strength using two-photon MNI-Glutamate uncaging prior to and 5, 15, and 30 min following 5-HT HFU at individual dendritic spines. Both MNI-Glutamate and RuBi-5-HT were uncaged by parking a two-photon beam at a point ~0.5 μm from the center of the spine head with a pulsed Ti:sapphire laser (MaiTai, Spectra-Physics). Shift 5-HT HFU was performed by uncaging RuBi-5-HT at a parallel distance from the dendritic shaft but >2 μm from the nearest dendritic spine. Spines exposed to shifted (2 μm away from uncaging spot) two-photon glutamate uncaging test pulses showed negligible uEPSCs. Experiments to assess changes in single-spine size, strength, and stability were carried out at 30 °C in ACSF containing (in mM) 2 $Ca^{2+}$, 1 $Mg^{2+}$, 0.001 TTX, 0.1 RuBi-5HT, and/or 2.5 MNI-Glutamate. To avoid confounds due to glutamate or 5-HT spillover, we chose unstimulated spines that were located at least 2 μm away from the nearest stimulated spine.

### Electrophysiology

Whole-cell recordings (electrode resistance, 5–8 MΩ; series resistance, 20–40 MΩ) were performed at 30 °C on visually identified layer 2/3 pyramidal neurons within 40 μm of the slice surface of dorsolateral PFC slice cultures (EP11-14) or acute slices (P6-24) using a MultiClamp 700B amplifier (Molecular Devices). Uncaging evoked excitatory postsynaptic currents (uEPSCs) were recorded in ACSF at 30 °C containing (in mM): 2 $Ca^{2+}$, 1 $Mg^{2+}$, 0.001 TTX, and 2.5 MNI-Glutamate. Neurons were patched in voltage-clamp configuration (Vhold = −65 mV) and uncaging test pulses (720 nm, 2 ms duration, 7–10 mW at the sample) at individual spines (50–100 μm from the soma on secondary and/or tertiary apical dendrites) elicited an average response of ~8 pA at the soma using cesium-based internal solution (in mM: 135 Cs-methanesulfonate, 10 HEPES, 10 $Na_2$ phosphocreatine, 4 $MgCl_2$, 4 $Na_2$-ATP, 0.4 Na-GTP, 3 Na L-ascorbate, 0.2 Alexa 488, ~300 mOsm, ~pH 7.25). uEPSC amplitudes were quantified as the average (5–8 test pulses at 0.1 Hz) from a 1 ms window centered on the maximum current amplitude within 50 ms after uncaging pulse delivery. When uEPSCs were measured from multiple spines within a region of interest, stimulated spines were located at least 2 μm away from one another to avoid confounds due to possible glutamate spillover. For single-spine two-photon 5-HT uncaging fLTP experiments, uEPSCs were acquired from one target and one unstimulated off-target spine on a separate branch at least 50 μm away. After a short baseline of uEPSCs, 5-HT HFU was delivered at the target spine while the cell continued to be held at −65 mV. Post 5-HT HFU uEPSCs were recorded at −65 mV from both target and off-target spines immediately after 5-HT stimulation and at 5–15 min intervals for 30 min following the 5-HT HFU protocol. Spontaneous action potential independent miniature excitatory postsynaptic currents (mEPSCs) were measured at 30 °C in voltage-clamp configuration (Vhold = −65 mV) using cesium-based internal solution in ACSF. mEPSC recordings ranged from 60 to 240 s. To determine the effect of short-term bath application and long-term in vivo administration of Clozapine N-Oxide (CNO) on neuronal excitability following DREADDs expression, resting membrane potentials and action potential firing rates were recorded in current-clamp mode using potassium-based internal solution (in mM: 136 K-gluconate, 10 HEPES, 17.5 KCl, 9 NaCl, 1 MgCl2, 4 Na2-ATP, 0.4 Na-GTP, and ~300 mOsm, ~pH 7.26) at 30 °C in ACSF containing 2 mM $Ca^{2+}$ and 1 mM $Mg^{2+}$. Signals were digitized at 10 kHz and responses were analyzed using Clampfit 10.3 (Molecular Devices) and OriginPro 8.5 (OriginLab) software.

### Quantification of densities and fluorescence intensities of dendritic spines and fluorescence intensities of GRAB 5-HT1.0

All distinct protrusions emanating from the dendritic shaft, regardless of shape, were counted and measured in images from the red (tdTomato in slice culture) or green (EGFP in slice culture or Alexa 488 in acute slice) channel using ImageJ (NIH). Estimated spine volume and SEP-GluA2 expression level were measured from background-

subtracted and bleed-through-corrected green (EGFP, Alexa 488, or SEP-GluA2) and red (tdTomato) fluorescence images using the integrated pixel intensity of an oval region (~1 μm²) surrounding the spine head in the single Z stack image in which the spine appeared the brightest[50,80]. The estimated spine size was calculated by normalizing the fluorescence intensities for individual spines on a single dendritic segment to the mean fluorescence intensities measured from four regions of interest (ROIs) on the dendritic shaft. Background subtraction for dendrites was performed by subtracting "adjacent background" (mean pixel intensity from 4 ROIs adjacent to the shaft multiplied by the number of pixels in the dendrite ROI) from the integrated dendritic ROI intensity. Background subtraction for spines was performed by subtracting "adjacent background" (integrated pixel intensity from a neighboring ROI of equal dimensions proximal to the dendritic shaft) from the integrated spine ROI intensity. For single-spine two-photon 5-HT uncaging experiments, target spines stimulated by 5-HT HFU or 5-HT LFU, all unstimulated off-target spines on the same region of interest, and all spines on far removed (>50 μm from 5-HT HFU or 5-HT LFU target) regions of interest were analyzed per cell. Baseline spine size was determined by averaging a series of two images 10–15 min apart for pre-existing spine sLTP experiments and 30 min apart for new spine stabilization experiments. Spines were categorized into small, medium, and large groups based on normalized fluorescent intensities (small: 0–60; medium: 60–150; large: >150)[81]. All spine size changes were measured as a percentage of average baseline fluorescence intensity[50,82,83]. Spines that increased by more than 3.5% from baseline measurements were considered to have undergone enlargement. Spine length/width ratio was defined as the ratio of the length from the tip of the spine head to the spine neck base (spine length) to the width across the spine head at its widest point (spine width) or, if the widest point was ambiguous, at the maximum dimension perpendicular to spine length[80]. Protrusions were classified as filopodia if the ratio of head width to neck width was <1.2 and the ratio of spine length to neck width was >3[84,85]. GRAB 5-HT two-photon imaging data was analyzed using ImageJ (NIH). The fluorescence changes (ΔF/F0) were calculated using the formula (F-F0) / F0, in which F0 is the baseline fluorescence signal. After measuring baseline fluorescence (F0, 0 min) from the cell body, fluorescence changes were measured (F, 30–40 min) after 5-HT or CNO bath application.

### Criteria for de novo formation and elimination of dendritic spines

For de novo spine formation, estimated spine size was calculated from red (tdTomato) or green (EGFP) signal intensities[78,80]. New spines were identified by imaging 4–6 sections of dendrites (ROIs) from secondary and/or tertiary apical dendrites 50–100 μm from the soma for a 30 min baseline period (−30 min and 0 min prior to spine identification). Spines were considered to have formed if a protrusion was present at 0 min that was not present at the first imaging time point at −30 min, and the intensity/pixel was >2 standard deviations (SDs) of the background signals near the dendrite, calculated from measuring the SD of 4 oval regions drawn adjacent to the dendrite and neighboring the new protrusion[78]. TTX was added immediately after new spines were identified. Spines were considered eliminated when the intensity/pixel dropped to <2 SDs of background levels near the dendrite and eliminated protrusion[80]. If there was uncertainty concerning the status of a new spine because of dendrite movement, swellings in the Z-axis, or neighboring spine movement, the spine was excluded.

### Intracranial injections

Adeno-associated viruses (AAVs) were injected using a hand-made stereotactic setup. Neonates (P0-1) were cryoanesthetized. Following cessation of movement, a solution of recombinant AAVs ($10^{12}$–$10^{13}$ GC/ml) was injected bilaterally into layer 2/3 of the dorsolateral PFC using a pulled glass needle (30–40 μm)[35]. The needle was held perpendicular

to the skull surface during insertion to a depth of approximately 200–300 µm. Once the needle was in place, viral solution (300–500 nL) was manually injected into each hemisphere (Micro-dispenser, VWR) using Retro-AAV-hSyn-DIO-hM3D(Gq)-mCherry, Retro-AAV-hSyn-DIO-hM4D(Gi)-mCherry (Addgene catalog #44361-AAVrg; #44362-AAVrg; gifts from Bryan Roth), Retro-AAV-FLEX-tdTomato (Addgene catalog #28306-AAVrg; gift from Edward Boyden), or AAV-hsyn-GRAB 5-HT1.0 (Addgene catalog #140552-AAV9; gift from Yulong Li). Litter, age, and gender-matched control animals received the same amount of vehicle solution. After injection, pups were placed on a warming pad for recovery ( < 15 min) and returned to the home cage[78]. All AAV and vehicle-treated animals were handled equally.

## Oral drug administration
All drugs for in vivo experiments were delivered by carefully handling pups and using a pipette to feed pups orally. Drugs were orally delivered at the following concentrations: Fluoxetine (10 µg/g starting from P2 up to P15, or starting from P16 up to P24), MDL 11939 (0.3 µg/g from P2-P6, 0.5 µg/g from P7), and DR 4485 hydrochloride (0.3 µg/g from P2-6, 0.5 µg/g from P7) once daily[55,56,86]. CNO (1 µg/g starting at P6) was administered twice daily[34–36]. Litter, age, and gender-matched controls received the same volume and concentration of vehicle (saline and/or DMSO). Both male and female mice were used for all in vivo, ex vivo, and slice culture experiments. Mouse weights were recorded on a small scale daily to monitor any changes in health compared to litter-matched controls. All drug and vehicle treated animals were handled equally.

## Pharmacology
Stocks were prepared at 1,000 x (or greater) by dissolving Tetrodo-toxin citrate, (R)-CPP, SB 269970 hydrochloride, TCB-2, and Clozapine N-oxide dihydrochloride in water; NBQX, MDL 100907, Gö 6983, SQ 22536, ML 141, 2c, MDL 11939, DR 4485 hydrochloride, and AS-19 in DMSO. All drugs were from Tocris unless otherwise noted. All drugs were applied at least 30 min prior to imaging, uncaging, and/or electrophysiological recordings. Vehicle controls were matched in identity and volume to that in which the drug was dissolved.

## Statistics and reproducibility
Data from a minimum of 3 independent slice culture preparations (average of 6) or a minimum of 3 mice per each group for acute slice and in vivo work were used for each experimental comparison. All statistics were calculated across cells. Error bars represent standard error of the mean and significance was set at $p = 0.05$ (Student's two-tailed $t$-test, non-parametric Mann-Whitney, or one way ANOVA). Survival rate across conditions for individual time points was compared by Fisher's exact test (two-sided) or using a log-rank test for survival comparisons throughout the entire imaging period[16,78]. Outliers were not excluded from the data. All data were included for analysis and figures. We declare that all data supporting the findings of this study are provided within the paper and its supplementary information. Single and double asterisks indicate $p < 0.05$ and $p < 0.01$, respectively; n.s., not significant.

## Reporting summary
Further information on research design is available in the Nature Portfolio Reporting Summary linked to this article.

# Data availability
All data needed to evaluate the conclusions in the paper are present in the paper and the Supplementary Materials. Source data generated in this study are provided in the Source Data file. Source data are provided with this paper.

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

## Acknowledgements

We thank Mark Dell'Acqua, Christopher Ford, and Matthew Kennedy for critical discussions; Hyunah Choo and Byungsun Jeon for compound 2c; Boram You for help with data analysis. Funding sources include T32NS099042 (R.O.); NICHD F31HD106632 (R.O.); NIH R01MH124778, R21MH126073, R21NS133681 (W.C.O.); Brain and Behavior Research Foundation (W.C.O.); Brain Research Foundation (W.C.O.); CSU/CU-Pilot Collaboration Award (W.C.O.); Ludeman Family Center Research Award (W.C.O.).

## Author contributions

Conceptualization: R.O. and W.C.O. Investigation: R.O., L.E.G.W., V.M.H., I.-W.H., V.N.C., W.C.O. Visualization: R.O. and W.C.O. Funding acquisition: R.O., and W.C.O. Supervision: W.C.O. Writing: R.O. and W.C.O. Editing: All authors.

## Competing interests

The authors declare no competing interests.
