## [Peer Review File · Nature Communications]

Serotonin modulates excitatory synapse maturation in the developing prefrontal cortexREVIEWER COMMENTS

Reviewer #1 (Remarks to the Author):

In Ogelman et al., the authors used chemogenetics, two-photon uncaging, and pharmacological manipulations to study how 5-HT signaling regulates dendritic spine density in developing PFC. I think it is one of the few studies in the field that studies how neuromodulators regulate synaptic plasticity using these detailed and elegant experimental approaches. The findings are also very novel, and I think they will generate interest to the general audience in the neuroscience community.

However, it is unclear why the manuscript is written in a short format, which really lacks some details on the background and rationale, clarity on the experimental methods, and interpretation of their findings. Below, I list some of the questions that were unclear to me.

Major:

1. 'DREADDs expressing pups showed normal feeding behavior'  what's the rationale for doing this experiment, and what's the interpretation?
2. Figure 2f and g  some other controls are necessary here. Since it's 2-color uncaging, there must be some worries about bleed through; hence, off-target stimulation doesn't seem to be a proper control. What does uEPSC look like when they just uncage 5-HT at the 30min mark? Can uncaging 5-HT itself evokes EPSC?
3. Following the same question above on glutamate and 5-HT, I'm a bit confused about glutamate-mediated LTP vs. 5-HT LTP now. How do the authors know if glutamate is not involved in the LTP induction when they just uncage 5-HT? Were the experiments done in a TTX and CPP condition? If that is the case, it should be clarified in the text because so far, only 'extended data fig 6' mentioned TTX and CPP. Also, is increased EPSC amplitude after 30min of stimulation a standardized readout for LTP? Additional evidence on 5HT LTP should be provided as this is a big claim of the manuscript.
4. The authors interchangeably used the terms 'LTP', 'sLTP', and 'fLTP' throughout the text. In my opinion, they really refer to different things and different measurements. If the authors are looking at the spine size, it should be sLTP. If the authors are looking at SEP-GluR or uEPSC, then it should be fLTP. They shouldn't use LTP to generalize everything. Also, please define sLTP and fLTP. I presume it's structural and functional?
5. Fig 3A  this part is very confusing. Is this following the Kwon protocol (Nature 2011) of uncaging Glu to induce new spines? Or are the authors claiming that uncaging 5HT alone can also induce new spines? If not, how do they know whether to stimulate and whether there will be a new spine form at that specific location?
6. In Fig 3h and I, 2AR blockade did not alter new spine survival but 7R blockade did, and the authors conclude that structural potentiation is not prerequisite for new spine stabilization \diamond the authors should test if the fLTP is required for new spine stabilization?
Also the authors never tested whether 2AR or 7R is required for fLTP?
7. The entire discussion needs to be expanded significantly. The authors should elaborate on what they think about how 2AR and 7R signaling are involved in sLTP and fLTP, and why they are so special during development. More importantly, how it is different than glutamate-induced LTP?

Minor:

1. In Ext Data Fig 1b and c  the authors showed an increase or decrease in membrane potential

after CNO application. But in the text, it says 'resulted in increased firing rates'. The authors should be more specific about what they quantified.

2. 5-HT High frequency stimulation (HFU)  why is 1 Hz a high frequency? In the LTP induction protocol, this would be considered as low frequency?

3. Please add line number to the manuscript so it makes it easier for the reviewers to refer to places.

Reviewer #2 (Remarks to the Author):

Ogelman et al. demonstrate the role of serotonin (5-HT) signaling in the developing prefrontal cortex. Chemogenetic suppression and enhancement of 5-HT from neurons projecting from raphe nuclei decreased and increased the density and strength of excitatory spine synapses of layer 2/3 pyramidal neurons, respectively, during the critical period. Serotonin uncaging experiments with serotonin receptor blockades show that spine LTP is induced by 5-HT_{2A}R and 5-HT₇R signaling, possibly through PKC activation downstream of 5-HT_{2A}R. The authors also showed that excitatory synapse stabilization is mediated by 5-HT₇R, independent of 5-HT_{2A}R. Lastly, they showed that 5-HT increase via fluoxetine treatment affects PFC excitatory synapses through 5-HT_{2A}R and 5-HT₇R signaling. Given the implicated roles of 5-HT and excitatory synapses in the regulation of brain development and function as well as various brain disorders, identifying key mechanisms linking these two major plays are very important. The experiments are well-designed and controlled, and the obtained data are largely convincing.

Major comments.

1. It is unclear whether the chemogenetic modulation of PFC-projecting raphe neurons indeed alters 5-HT release in the target regions is unclear. The authors could use GRAB sensors or some other methods (i.e. dialysis) to demonstrate that.

2. Does 5-HT modulation have similar effects on layer 5 neurons? Given the different roles of superficial- and deep-layer cortical neurons, demonstration of similar effects in the two different layers would much strengthen the manuscript.

Minor comments

1. The authors focus on 5-HT_{2A}R and 5-HT₇R, but the reason for this focus is unclear. Are other types of 5-HT receptors not expressed in the PFC? Are 5-HT_{2A}R and 5-HT₇R expressed in the layer 2/3 of PFC at early postnatal stages?

2. In Figure 4, 5-HT receptor antagonists only have no effect on synaptogenesis. However, the working model of the present study predicts decreases in excitatory synapse formation and maturation by 5-HT receptor antagonists. This should be discussed.

3. There seem to be some variations in the WT values in chemogenetic activation and inhibition in Figure 1. This should be explained.

4. The authors mention that gender-matched mice were used. Since serotonin has been reported to have different effects on males and females, I wonder whether there are any gender differences in the current results.

5. Extended Data Figure 1 tests the expression and effect of rAAV-hM4D/hM3D at P11-15. However, Figure 1a shows the effects of CNO treatment that starts at P6. Is there any reason for the difference?

6. What could be the reason for the difference between the lack of changes in spine size in fluoxetine and DREADD experiments (Figures 1 and 4) and the positive changes in spine size in 5-HT uncaging experiments (Figure 2e)?

7. The authors suggested two separate mechanisms for synaptic potentiation in mature and newly-formed spines, involving 5-HT₇R in both cases, and requiring additional 5-HT_{2A} in the case of mature spine maintenance. Could they provide their hypotheses behind the need for distinctive mechanisms in synaptic maintenance and potentiation between old and new synapses in their discussion? Why an additional receptor is required for mature synapses?

8. I wonder if the fluoxetine treatment at late (i.e. adult) stages has no effect on excitatory synapses. Although this could be asking the authors too much, addressing this question could support the idea that the early postnatal critical period is important.

Reviewer #3 (Remarks to the Author):

Remarks to the Authors:

In this study, Ogelman et al. explored the effects of serotonin on layer 2/3 pyramidal neurons in the prefrontal cortex, during early development. First, the authors showed that chronic chemogenetic activation and inhibition of SERT-positive neurons increased and decreased spine density of layer 2/3 neurons in the PFC. These results correlate with those obtained with chronic fluoxetine administration, which is known to increase serotonin level and alter spine density. Using an elegant approach of serotonin uncaging, the authors demonstrated that serotonin induces both structural and functional LTP through the activation of 5-HT_{2A}R and 5HT₇R. In contrast, new spine stabilization requires 5-HT₇R activation selectively. These findings are important in the field as they highlight the critical role of serotonin as a mediator of PFC development.

Overall, the experiments are well designed, and the results are potentially significant and support most of the claims that are made. However, there are several important loose ends the authors should address, and additional control experiments should be performed. The discussion is poorly written as the authors summarized their findings, but almost did not mention current literature supporting/challenging their results. Finally, the manuscript will benefit from a schematic diagram highlighting the contribution of both receptors (i.e., localization, role, signaling).

MAJOR COMMENTS:

1) Throughout the paper, the authors analyzed spine density and spine stabilization in layer 2/3 pyramidal neurons of the PFC. Spines are classified as small, medium and large relative to the fluorescence intensity. It is unclear how this criterion has been defined. A deeper analysis should be performed to evaluate the effect of serotonin on mature versus immature spines, using the standard classification of spine shape (e.g., filopodium, thin, stubby, mushroom). Moreover, the authors did not mention whether the analysis was performed on basal and/or apical dendrites and the potential differences, as these spines receive different inputs during development. All these details should be taken into consideration in order to interpret the results on both sLTP and fLTP.

2) In Figure 1, the authors attempted to test the effect of chronic serotonin release. The authors used inhibitory and excitatory DREADDs to increase and decrease the activity of SERT-positive neurons infected in the dorsal raphe, respectively. The authors confirmed that bath application of CNO decreases and enhances the firing of hM4D(Gi)- or hM3D(Gq)-expressing DRN neurons, respectively.

However, the authors never showed the expression of DREADDs in the DRN (widefield image), their colocalization with SERT, and the expression in serotonergic fibers that innervate the PFC. More importantly, the authors should test the effect of acute CNO application in the PFC (as in Extended Data 1) because their current manipulation is not specific for the serotonergic fibers that innervate the PFC. Finally, the authors used CNO to activate DREADDs, however the observed effects could arise from back-metabolism to clozapine or binding to endogenous neurotransmitter receptors, including 5-HT_{2A} and 5-HT₇ receptors. As an alternative, the authors should consider using a different DREADD agonist (e.g., DREADD agonist 21) or optogenetic stimulation of SERT-positive fibers originated from DRN (Hyun et al. 2023 PMID: 36945634).

3) The authors used 5-HT uncaging to test the acute effect of serotonin at the spine level. However, some clarifications and improvements are required to fully appreciate and clearly interpret the results. All data looking at spine size upon 5-HT uncaging should show the baseline (i.e., before 5-HT HFU). Moreover, it is unclear how the authors determined the protocol of 5-HT HFU. Does it mimic physiological release of serotonin at this developmental stage?

4) The authors provided evidence that serotonin can mediate spine stabilization through 5-HT_{7R} activation, but serotonin also induces sLTP and fLTP through the activation of 5-HT_{2ARs} and 5-HT_{7Rs}. However, the results rely entirely on pharmacology, and the 5-HT_{2AR} and 5-HT_{7R} antagonists may not be very selective. It is necessary that the authors use more than one drug to block 5-HT_{2ARs} and 5-HT_{7Rs}. They should also consider using knockout mice for these receptors. Moreover, it is important to know whether selective activation of 5-HT_{2ARs} and/or 5-HT_{7Rs} (using selective agonists) mimics these effects.

5) While activation of DRN neurons using hM3D(Gq) increases spine density in layer 2/3 pyramidal neurons of the PFC, the authors showed that uEPSCs are also enhanced. Which kind of spine were targeted for uncaging (i.e., mature or immature)? uEPSC decays look different (e.g., for the traces showed in Fig. 1f and 1j).

6) In Figure 4, the authors treated fluoxetine mice chronically with 5-HT_{2AR} and 5-HT_{7R} antagonists. What is the effect of these antagonists administered alone?

7) The authors focus on 5-HT_{2ARs} and 5-HT_{7Rs}. However, other serotonin receptors are present in the PFC at this development stage and may be critical for spine maturation.

8) In the abstract, the authors mentioned the effect of serotonin "during the critical period". However, they never defined it and they never discussed it. Are these effects different later during PFC maturation or adulthood?

MINOR COMMENTS:

1) For experiments measuring uEPSCs, the authors should show the individual spines, not only the mean.

2) The authors should report whether chronic manipulations, such as DREADDs and fluoxetine, influence dendritic arborization and complexification (Sholl analysis) as well as intrinsic properties of layer 2/3 pyramidal neurons.

3) The authors did not mention whether they use males and/or females.

4) The authors did not mention which part of the PFC they focused on.

5) The authors may want to revise the title as the study focuses on the effect of serotonin on dendritic spine during PFC development.

MS# NCOMMS-23-17520-T

Reviewer Comments:

Reviewer #1

We thank Reviewer 1 for their thoughtful comments and suggestions. We appreciate the reviewer's remarks that highlight our use of detailed and elegant approaches and emphasize that our findings are very novel as there are few studies in the field that examine neuromodulatory regulation of synaptic plasticity. All the comments and concerns are thoroughly addressed in our revised manuscript and were greatly welcomed as they have significantly improved the quality of our findings. We now include important uncaging controls to show that there is no bleed-through from uncaging laser wavelengths. We repeated 2-photon 5-HT HFU structural LTP experiments in the presence of TTX, NBQX, and CPP to further rule out glutamatergic contributions to structural LTP. We have made the appropriate changes to our manuscript to clarify the differences between structural LTP and functional LTP and now demonstrate the requirement of 5-HT_{2A}R and 5-HT₇R in functional LTP to delineate similarities between the two mechanisms. Below we respond to each of the comments.

Comments:

In Ogelman et al., the authors used chemogenetics, two-photon uncaging, and pharmacological manipulations to study how 5-HT signaling regulates dendritic spine density in developing PFC. I think it is one of the few studies in the field that studies how neuromodulators regulate synaptic plasticity using these detailed and elegant experimental approaches. The findings are also very novel, and I think they will generate interest to the general audience in the neuroscience community.

However, it is unclear why the manuscript is written in a short format, which really lacks some details on the background and rationale, clarity on the experimental methods, and interpretation of their findings. Below, I list some of the questions that were unclear to me.

Major comments:

1.1) DREADDs expressing pups showed normal feeding behavior  what's the rationale for doing this experiment, and what's the interpretation?

On the original manuscript, we reported feeding behavior by measuring pup weights daily from the start of CNO administration until experiments were performed. As 5-HT_{2A}R disbalances have been shown to affect hunger, feeding habits, and weight gain (PMIDs: 1728826, 34290371, 36733453, 37525500), we wanted to ensure that changes in excitatory synaptic properties were not attributed to altered physical status in DREADD injected, CNO treated mice (now **Supplementary Fig. 8**). We have now added additional weight comparisons from CNO only vs saline treated control mice to show that oral CNO administration alone does not affect the health and weight of mice during this developmental period.

We have clarified this point in the results section of the revised manuscript (lines 116-118). This new weight comparison data set is included in Supplementary Fig. 7.

1.2) Figure 2f and g  some other controls are necessary here. Since it's 2-color uncaging, there must be some worries about bleed through; hence, off-target stimulation doesn't seem to be a proper control. What does uEPSC look like when they just uncage 5-HT at the 30min mark? Can uncaging 5-HT itself evokes EPSC?

...worries about bleed through

We appreciate this critical input. Although MNI caged compounds photolyze at 720 nm but not 810 nm wavelength 2-photon light (PMIDs: 30687075, 31354469; unpublished data in the previous labs of the PI), to address this concern specifically using our 2-photon system, we performed uncaging bleed through control experiments with MNI-glutamate (2.5 mM) in ACSF (2 mM Ca²⁺, 1 mM Mg²⁺) and compared uncaging-evoked EPSCs (uEPSCs) recorded at -65 mV (5-8 uncaging pulses at 0.1 Hz) tuned to either 720 nm or 810 nm wavelength. Importantly, uEPSCs were measured from the same spines by uncaging at 720 nm or at 810 nm. All spines showed clear uEPSCs at 720 nm, but uncaging responses were identical to off-stim responses when uncaging MNI-glutamate at 810 nm.

These new results are included in Supplementary Fig. 14a, b.

Can uncaging 5-HT itself evokes EPSC?

We thank Reviewer 1 for this concern, as 5-HT₃ receptor activation can induce rapid ionotropic inward currents (PMIDs: 17073663, 9356400). To address this concern, we uncaged RuBi-5-HT using 810 nm wavelength pulses while recording uEPSCs at -65 mV. No detectable responses were measured as compared to baseline.

These new results are added in Supplementary Fig. 14c, d.

Together, these data demonstrate that during the 5-HT HFU protocol (30 pulses of 1 ms duration at 1Hz by 810 nm) glutamate was not released and uEPSCs recorded at -65 mV were not evoked by 5-HT.

1.3) Following the same question above on glutamate and 5-HT, I'm a bit confused about glutamate-mediated LTP vs. 5-HT LTP now. How do the authors know if glutamate is not involved in the LTP induction when they just uncage 5-HT? Were the experiments done in a TTX and CPP condition? If that is the case, it should be clarified in the text because so far, only 'Supplementary fig 6' mentioned TTX and CPP. Also, is increased EPSC amplitude after 30min of stimulation a standardized readout for LTP? Additional evidence on 5HT LTP should be provided as this is a big claim of the manuscript.

How do the authors know if glutamate is not involved in the LTP induction when they just uncage 5-HT?

We thank Reviewer 1 for this important comment. As mentioned, 'Supplementary fig 6' in our original manuscript employs *1-photon* uncaging of 5-HT with TTX, CPP, and NBQX in ACSF, successfully inducing sLTP of dendritic spines (now **Supplementary Fig. 12a-d**). While these are critical data demonstrating that 5-HT can induce sLTP independently of glutamatergic signaling, our *2-photon* 5-HT uncaging experiments on the original manuscript were done in TTX but not CPP or NBQX. In this scheme, action potential-dependent neurotransmitter release was abolished. Additionally, in the originally submitted manuscript we showed that 5-HT_{2A} and 5-HT₇ receptor antagonists prevented sLTP (now **Figs. 3, 4**), supporting our finding that 5-HT induces sLTP through 5-HTR activation rather than glutamatergic receptor activation. However, to further confirm our findings, in our revised manuscript we have added new 5-HT HFU experiments with the addition of TTX, CPP, and NBQX in ACSF. We observed a comparable magnitude of sLTP to experiments done in TTX only ACSF, demonstrating that 5-HT can induce sLTP independently of glutamatergic signaling. In addition, we verified that glutamate is not photo-released by 810 nm, as shown in **Supplementary Fig. 14**.

These new data are added in Supplementary Fig. 12e, f.

Is increased EPSC amplitude after 30min of stimulation a standardized readout for LTP

Yes, 30 min readouts of LTP are a standard readout of LTP. Please see the following references: PMIDs, 23303946, 36395772, 30643148, 18556515, 23269840

1.4) The authors interchangeably used the terms ‘LTP’, ‘sLTP’, and ‘fLTP’ throughout the text. In my opinion, they really refer to different things and different measurements. If the authors are looking at the spine size, it should be sLTP. If the authors are looking at SEP-GluR or uEPSC, then it should be fLTP. They shouldn’t use LTP to generalize everything. Also, please define sLTP and fLTP. I presume it’s structural and functional?

Thank you to the Reviewer 1 for pointing this out. Yes, sLTP refers to *structural* long-term potentiation and fLTP refers to *functional* long-term potentiation. We agree that sLTP and fLTP have different physiological meanings and mechanisms, although activity-dependent sLTP is positively correlated with fLTP (PMIDs: 15190253, 25558061, 23269840, 23303946); our revised manuscript differentiates between the two much more clearly.

1.5) Fig 3A  this part is very confusing. Is this following the Kwon protocol (Nature 2011) of uncaging Glu to induce new spines? Or are the authors claiming that uncaging 5HT alone can also induce new spines? If not, how do they know whether to stimulate and whether there will be a new spine form at that specific location?

We appreciate the reviewer’s concern. New spines were identified by imaging 4-6 sections of dendrites (regions of interest, ROIs) for a 30 min baseline period to examine if any new spines were spontaneously formed during this timeframe (**Fig. 5a**). Although we did not know whether there would be new spines in any ROI, several ROIs often had a new spine naturally formed in an activity-dependent manner during the 30 min baseline. Once we identified nascent spines, we stimulated a ‘target’ new spine by 5-HT HFU to examine post-formation dynamics and survivorship. 1-3 new spines per cell were not stimulated and considered ‘off-target’ control new spines. Importantly, we added TTX immediately after new spines were identified. Therefore, both target and off-target spines were formed in an activity-dependent manner and only ‘target’ spines were exposed to 5-HT in the absence of neuronal activity, while ‘off-target’ spines served as internal controls.

We have clarified the new spine stabilization protocol in our methods of the revised manuscript (lines 693-705).

1.6) In Fig 3h and I, 2AR blockade did not alter new spine survival but 7R blockade did, and the authors conclude that structural potentiation is not prerequisite for new spine stabilization. The authors should test if the fLTP is required for new spine stabilization? Also the authors never tested whether 2AR or 7R is required for fLTP?

The authors should test if the fLTP is required for new spine stabilization

We fully agree with Reviewer 1. To address this question, we performed a new set of experiments to determine if nascent spines exposed to 5-HT HFU undergo fLTP. We monitored surface AMPARs in new spines by imaging SEP-GluA2 transfected layer 2/3 pyramidal neurons and found that 5-HT HFU increased surface AMPAR expression of *survived* new spines by 30 min post stimulation, demonstrating that newly formed spines stabilized by 5-HT HFU undergo fLTP. Importantly, we found that *eliminated* new spines did not increase in SEP-GluA2 expression at 30 min post-formation, strongly suggesting that both fLTP and sLTP (now **Fig. 5a-f**) are critical for 5-HT induced new spine stabilization.

These new results are shown in Fig. 5e, f.

authors never tested whether 2AR or 7R is required for fLTP?

To test the requirement of 5-HT_{2A}R and 5-HT₇R in fLTP, we employed SEP-GluA2 imaging and performed 5-HT HFU in the presence of 5-HT_{2A}R and 5-HT₇R antagonists. We found that surface AMPARs were not increased following 5-HT HFU stimulation, indicating that 5-HT_{2A}R and 5-HT₇R are required for fLTP.

These new results are shown in Fig. 4g, h and Supplementary Table 3.

The requirement of 5-HT_{2A}R and 5-HT₇R in sLTP is now shown in Fig. 3.

In addition, we repeated sLTP experiments using alternative antagonists for 5-HT_{2A}R (MDL11939, 1 μ M) and 5-HT₇R (DR4485 hydrochloride, 5 μ M), which are the same antagonists we used for our *in vivo* oral administration experiments (Fig. 3f, g).

1.7) The entire discussion needs to be expanded significantly. The authors should elaborate on what they think about how 2AR and 7R signaling are involved in sLTP and fLTP, and why they are so special during development. More importantly, how it is different than glutamate-induced LTP?

We agree with Reviewer 1. Because our original manuscript was automatically transferred from another journal, which requires a brief format, to the current *Nature Communications*, our discussion indeed needs more details on 5-HTergic receptor signaling during development.

We have revised the entire discussion section.

Minor comments:

1.8) In Ext Data Fig 1b and c  the authors showed an increase or decrease in membrane potential after CNO application. But in the text, it says ‘resulted in increased firing rates’. The authors should be more specific about what they quantified.

We thank Reviewer 1 for pointing out the confusion in ‘Ext Data Fig 1b and c’ of our original submission. We have clarified our quantification in the revised manuscript with additional experiments (1) to explain that we measured resting membrane potentials of hM3D(Gq) or hM4D(Gi) expressing 5-HTergic neurons on brainstem slices that contain dorsal raphe nucleus (DRN) of SERT-Cre mice and (2) to demonstrate that CNO bath application onto hM3D(Gq) or hM4D(Gi) expressing DRN containing brainstem slices decreases and increases rheobase for action potential firing, respectively. We have also increased the range of ages for these experiments from P11-15 in our original manuscript to P6-15 in the revised manuscript to better mimic the ages of oral CNO administration *in vivo*. Importantly, vehicle (water) treated control slices were age and gender-matched, and we found that there are no differences in firing property changes between the two sexes.

We have clarified this point in the results section of the revised manuscript (lines 83-89).

These new results are added in Supplementary Fig. 1f-j.

1.9) 5-HT High frequency stimulation (HFU)  why is 1 Hz a high frequency? In the LTP induction protocol, this would be considered as low frequency?

High Frequency *Uncaging* stimulation (HFU) has been defined at 1 Hz in several published works involving glutamatergic activity-dependent LTP (PMIDs: 36395772, 30643148, 28402855). In conventional *electrical* stimulation protocols, 1 Hz is considered low frequency to induce long-

term depression (LTD) (PMIDs: 1350090, 15572107, 23269840), but to account for the low release probability of glutamate, 0.1 Hz is used as a low frequency protocol for LTD induction in glutamate *uncaging* protocols (PMIDs: 23269840, 26338340).

We originally chose 1 Hz as our 5-HT uncaging stimuli because it is within the physiological range of 5-HTergic neuronal firing (PMIDs: 27536220, 22076606). In our revised manuscript we performed our 2-photon 5-HT uncaging sLTP experiment using a low frequency 0.1 Hz protocol with the same number of pulses, which we are defining as Low Frequency Uncaging stimulation (5-HT LFU, 30 pulses of 1 ms duration at 0.1 Hz). 5-HT LFU did not induce sLTP, supporting our claim that 1 Hz is a higher LTP inducing frequency as compared to 0.1 Hz and highlighting the importance of pattern specificity for 5-HTergic activity in sLTP.

This new data set is added in Fig. 2e.

1.10) Please add line number to the manuscript so it makes it easier for the reviewers to refer to places.

We thank Reviewer 1 for this suggestion. **Line numbers are added in the revised manuscript.**

Reviewer #2:

We thank Reviewer 2 for their positive remarks about our study and appreciate that they found these experiments to be well-designed, controlled, and convincing. All the suggested experiments have been addressed thoroughly and have been very helpful in improving the quality of our manuscript. As suggested, we have added GRAB-5-HT1.0 control experiments to confirm that CNO application releases 5-HT in the PFC of hM3D(Gq)-DREADDs expressing SERT-Cre mice. To identify any layer specificity in our results, we tested whether 5-HT HFU could induce structural long-term potentiation (sLTP) on layer 5 pyramidal neurons. We also examined the effect of fluoxetine on older pups at P16-24. Additionally, we have expanded our discussion section as an article format.

Comments:

Ogelman et al. demonstrate the role of serotonin (5-HT) signaling in the developing prefrontal cortex. Chemogenetic suppression and enhancement of 5-HT from neurons projecting from raphe nuclei decreased and increased the density and strength of excitatory spine synapses of layer 2/3 pyramidal neurons, respectively, during the critical period. Serotonin uncaging experiments with serotonin receptor blockades show that spine LTP is induced by 5-HT_{2A}R and 5-HT₇R signaling, possibly through PKC activation downstream of 5-HT_{2A}R. The authors also showed that excitatory synapse stabilization is mediated by 5-HT₇R, independent of 5-HT_{2A}R. Lastly, they showed that 5-HT increase via fluoxetine treatment affects PFC excitatory synapses through 5-HT_{2A}R and 5-HT₇R signaling. Given the implicated roles of 5-HT and excitatory synapses in the regulation of brain development and function as well as various brain disorders, identifying key mechanisms linking these two major plays are very important. The experiments are well-designed and controlled, and the obtained data are largely convincing.

Major comments:

1.1) It is unclear whether the chemogenetic modulation of PFC-projecting raphe neurons indeed alters 5-HT release in the target regions is unclear. The authors could use GRAB sensors or some other methods (i.e. dialysis) to demonstrate that.

We fully agree with Reviewer 2 and appreciate this critical comment. We performed a number of new experiments to demonstrate that chemogenetic modulation of PFC-projecting raphe neurons in SERT-Cre mice alters 5-HT release in the PFC.

AAV-GRAB 5-HT1.0 (addgene: 140552; PMID: 33821000) was used, as Reviewer 2 suggested. First, to verify GRAB 5-HT sensor in our system, we injected AAV-GRAB 5-HT1.0 into the PFC of SERT-Cre mice at P1 and acute PFC slices were made at P14-15. Using 2-photon microscopy, we confirmed that GRAB 5-HT fluorescence is increased by bath application of 5-HT (30 min, 10 μ M; PMID: 33821000). We then co-injected AAV-GRAB 5-HT1.0 and retroAAV(rAAV)-DIO-hM3D(Gq) into the PFC of SERT-Cre at P1 and examined GRAB 5-HT fluorescence changes in acute PFC slices at P14-15 after CNO bath application (30-40 min, 1 μ M). We found that CNO bath application significantly increased GRAB fluorescence. To ensure CNO alone does not induce off-target increases in 5-HT sensor fluorescence, we measured GRAB 5-HT fluorescence changes following CNO bath application in the PFC of non-hM3D(Gq) injected mice. We also measured spontaneous changes in GRAB 5-HT fluorescence in acute PFC slices without CNO treatment. Neither caused any significant changes in GRAB 5-HT fluorescence. These experiments confirm that chemogenetic enhancement of PFC-projecting 5-HT neurons increases 5-HT release in the PFC.

These new results are shown in Supplementary Fig. 6.

Additionally, we functionally verified DREADDs by measuring mEPSCs from prefrontal layer 2/3 pyramidal neurons after CNO bath application in hM3D(Gq) expressing mice. CNO bath application (45-90 min, 1 μ M) induced a rightward shift in mEPSC amplitude. A similar increase in mEPSC amplitude was observed in 5-HT treated PFC slices (60 min, 10 μ M; shown in the original manuscript) and 5-HT_{2A}R and 5-HT₇R agonists treated PFC slices (60-90 min, TCB-2, 10 μ M; AS-19, 5 μ M).

These new and original results are included in Supplementary Figs. 10, 11 and 15.

Within the dorsal raphe nucleus (DRN), we measured resting membrane potentials and rheobase for action potential firing of hM3D(Gq) or hM4D(Gi) expressing 5-HTergic neurons in brainstem slices that contain DRN of SERT-Cre mice. Both hM3D(Gq) and hM4D(Gi) were confirmed by CNO bath application (30-60 mins, 1 μ M). In addition, we have increased the range of ages for these experiments from P11-15 in our original manuscript to P6-15 in the revised manuscript to better mimic the ages of oral CNO administration *in vivo*. Importantly, vehicle (water) treated control slices were age and gender-matched, and we found that there are no differences in firing property changes between the two sexes.

These new results are added in Supplementary Fig. 1f-j.

1.2) Does 5-HT modulation have similar effects on layer 5 neurons? Given the different roles of superficial- and deep-layer cortical neurons, demonstration of similar effects in the two different layers would much strengthen the manuscript.

We thank Reviewer 2 for suggesting this interesting experiment that strengthened our manuscript. We repeated 5-HT HFU experiments on dendritic spines of layer 5 pyramidal neurons. Layer 5 in the PFC has also been known to express both 5-HT_{2A}Rs and 5-HT₇Rs (PMIDs: 9435262, 29033796, 10462127), so it seemed plausible that 5-HT HFU may be able to induce sLTP. Surprisingly, 5-HT HFU onto layer 5 excitatory synapses did not induce sLTP. We have discussed this point in the discussion of the revised manuscript (**lines 375-378**).

These new results are included in Fig. 2e.

Minor comments:

1.3) The authors focus on 5-HT_{2A}R and 5-HT₇R, but the reason for this focus is unclear. Are other types of 5-HT receptors not expressed in the PFC? Are 5-HT_{2A}R and 5-HT₇R expressed in the layer 2/3 of PFC at early postnatal stages?

We chose 5-HT_{2A} and 5-HT₇ receptors to target for our experiments for several reasons that are now clearly stated in our revised manuscript. Although several subtypes of 5-HT receptors are expressed in the PFC, we identified 5-HT_{2A}Rs and 5-HT₇Rs as highly enriched in the early postnatal developing PFC and located post-synaptically (PMIDs: 9435262, 29033796, 10462127, 19889983). Past studies showed that pyramidal neuron depolarization by 5-HT in the PFC during early postnatal development is induced by a combination of 5-HT_{2A}R and 5-HT₇R activation (PMIDs: 15152041, 14742723, 17535909). Furthermore, 5-HT_{2A}Rs and 5-HT₇Rs are excitatory receptors coupled to either G α_q or G α_s downstream signals that are known to upregulate synaptic plasticity (PMIDs: 22378867, 10462127, 27013076, 19889983, 36030255). Since we observed spine density increases, spine strengthening, and spine enlargement in response to increased 5-HT by chemogenetics and 5-HT uncaging, and molecules downstream of G α_q or G α_s activation are critical for spine stabilization and growth (PMIDs: 25855176, 28737723, 32105741), we chose 5-HT_{2A}R and 5-HT₇Rs to target for our pharmacological experiments.

We have clarified this point in the introduction, results, and discussion sections of the revised manuscript (lines 59-66, 176-177, 180-182, 348-375).

1.4) In Figure 4, 5-HT receptor antagonists only have no effect on synaptogenesis. However, the working model of the present study predicts decreases in excitatory synapse formation and maturation by 5-HT receptor antagonists. This should be discussed.

We agree with Reviewer 2 that this was a surprising result, but there are several interpretations that could explain why our working model is different from what may have been expected from *in vivo* antagonist experiments. Our working model is specific to 5-HT post-synaptic activation *within* the PFC. Antagonists were administered orally, thus reaching the *entire* central nervous system, and inhibiting 5-HT_{2A}R and 5-HT₇R in many locations that may innervate and modulate the PFC development. The circuit wide effects of oral administration of 5-HT_{2A} and 5-HT₇R antagonists could differ from local inhibition of 5-HT_{2A}R and 5-HT₇R within the PFC at the synapse level. Furthermore, in our antagonists only experiments, drugs were only administered once per day which may have not been sufficient to block enough 5-HT_{2A} and 7 receptor activity to alter spine density or strength. Please note that in the *fluoxetine + antagonists* experiments, antagonists were administered 10 to 20 mins prior to fluoxetine delivery. Future follow up experiments could examine if more frequent antagonist treatments or knockdown of specific 5-HT receptors such as 5-HT_{2A} and/or 5-HT₇R in layer 2/3 pyramidal neurons of the PFC would induce the decrease in spine density and strength that our model predicts.

We have revised the discussion to address this point (lines 407-417).

1.5) There seem to be some variations in the WT values in chemogenetic activation and inhibition in Figure 1. This should be explained.

The reason for variations between controls of hM3D(Gq) and hM4D(Gi) chemogenetic experimental data is due to slight variations in the ages of mice for experiments. hM4D(Gi) injected mice had a higher proportion of more mature mice aged between P13-15 than slightly younger mice aged P11-12, as compared to hM3D(Gq) DREADDs mice. Since there is a rapid increase in synapse density in early postnatal development (PMIDs: 18048342, 30911152, 33795678), these slight variations in experimental ages account for the differences in spine density of controls in DREADDs experiments. As we explained in our original manuscript, due to a dramatic increase in spine density between P11 and P15 (P11: 0.85 +/- 0.05 spines/ μ m, 15 cells; P15: 1.35 +/- 0.05 spines/ μ m, 21 cells; $P < 0.01$), all CNO only controls in hM3D(Gq) and hM4D(Gi) chemogenetic experiments were age, litter, and gender-matched to DREADDs mice. In addition, we performed 2-photon imaging, uncaging, and electrophysiology experiments from both control and DREADD slices on the same day using the same batch of caged compounds, the same solutions including ACSF and internal solutions, and the same imaging and uncaging laser power to minimize non-biological variables.

We have clarified this point in the results and methods section of the revised manuscript (lines 92-95, 727-731).

1.6) The authors mention that gender-matched mice were used. Since serotonin has been reported to have different effects on males and females, I wonder whether there are any gender differences in the current results.

We appreciate Reviewer 2's question about sex differences in our results. Several behavioral studies have shown sex differences in 5-HT_{2A} signaling (PMIDs: 30430940, 35963476, 35904324, 35183539), so this is an important point. In the revised manuscript, we compare and report resting membrane potentials of hM3D(Gq) and hM4D(Gi) expressing 5-HT_{2A} neurons in

the raphe nuclei for males and females after CNO treatment but found no sex-differences between the two groups. We also found a comparable increase in the amplitude of mEPSCs between male and female mice induced by CNO-mediated 5-HT release in the PFC. For PFC slice culture work, we made slice cultures using both male and female pups at P3-4 and slices were randomly used for experiments at EP11-14 [equivalent postnatal (EP) day = postnatal day at slice culturing + days *in vitro*]. We thus cannot conclude whether there are sex differences in slice culture data. Future studies looking further into whether there are any age-dependent sex effects will be critical next steps in this work.

We have added new data in Supplementary Fig. 1h, j.

We also revised the results and discussion sections to address gender differences in our work (lines 135-140, 144-145, 382-389).

1.7) Supplementary Figure 1 tests the expression and effect of rAAV-hM4D/hM3D at P11-15. However, Figure 1a shows the effects of CNO treatment that starts at P6. Is there any reason for the difference?

We thank Reviewer 2 for identifying this concern. We repeated DREADDs verification experiments using younger pups and our revised manuscript now includes data from P6 pups, thus better mimicking the ages of oral CNO administration *in vivo* (CNO from P6-15). CNO bath application had the similar effect on 5-HTergic neurons at P6 as it did at P11-15, with CNO in hM3D(Gq) expressing slices increasing (depolarizing) resting membrane potential (RMP) and decreasing rheobase, and CNO onto hM4D(Gi) expressing neurons showing decreased RMP and increased rheobase.

These new results are included in Supplementary Fig. 1f-j.

1.8) What could be the reason for the difference between the lack of changes in spine size in fluoxetine and DREADD experiments (Figures 1 and 4) and the positive changes in spine size in 5-HT uncaging experiments (Figure 2e)?

There are several reasons that could explain the differences in spine sizes observed between our *in vivo* models and our 5-HT uncaging experiments. While we *did* observe a rightward shift in cumulative distribution of spine sizes in our hM3D(Gq) expressing mice (now **Supplementary Fig. 2c, d**), the lack of increase in spine sizes in fluoxetine exposed mice (now **Fig. 6c and Supplementary Fig. 16a, b**) is an interesting result, because of the positive changes in spine size in 5-HT uncaging experiments (**Figs. 2, 4**). Spine size change discrepancies between *in vivo* fluoxetine work and PFC slice work could be due to differences in circuit and cellular level 5-HT vs effects of 5-HT at the level of individual spines. In our *in vivo* fluoxetine models, the effect of fluoxetine cannot be confined to excitatory post-synapses, and thus we lack the ability to target post-synaptic 5-HT receptors specifically in layer 2/3 pyramidal neurons in the PFC. Additionally, the long term (chronic manipulations) vs short term (uncaging manipulation) effects of 5-HT may vary. While we did not identify changes in spine size following prolonged fluoxetine exposure, we observed post-synaptic strengthening of excitatory synapses in our *in vivo* fluoxetine models (**Fig. 6e**). These results suggest that while the 5-HTergic mechanisms underlying spine size changes are not sustained over these longer time periods, glutamatergic receptor (i.e. AMPAR) composition is altered, either through changing receptor properties, packing AMPARs in more densely, and/or possibly even rearranging nanocolumns (PMIDs: 28521126, 27080385, 27462810, 29096080).

We have discussed this point in the revised manuscript (lines 390-406).

1.9) The authors suggested two separate mechanisms for synaptic potentiation in mature and newly-formed spines, involving 5-HT₇R in both cases, and requiring additional 5-HT_{2A} in the case of

mature spine maintenance. Could they provide their hypotheses behind the need for distinctive mechanisms in synaptic maintenance and potentiation between old and new synapses in their discussion? Why an additional receptor is required for mature synapses?

This is another excellent point. We have expanded our discussion significantly to address 5-HT7R and 5-HT2AR dependent pathways and their involvement in spine maintenance and potentiation. 5-HT7Rs signal through G_{α_s} coupled proteins, which activate cAMP-dependent voltage gated Ca^{2+} channels (VGCCs) and increase extracellular Ca^{2+} entry into spines. In newly formed spines, the increase in Ca^{2+} through 5-HT7R dependent mechanisms may be sufficient to increase stabilization (PMIDs: 25498985, 18667153, 21295598, 23303946, 22334212). 5-HT2ARs signal through G_{α_q} coupled proteins, notably activating PKC. We believe that during 5-HT induced LTP, increased Ca^{2+} entry through 5-HT7R VGCC activation contributes to 5-HT2AR dependent PKC activation, as PKC requires Ca^{2+} to be activated (PMIDs: 30013171, 30013172, 2479143) during synaptic potentiation.

We have revised the discussion to emphasize this point (lines 335-361).

1.10) I wonder if the fluoxetine treatment at late (i.e. adult) stages has no effect on excitatory synapses. Although this could be asking the authors too much, addressing this question could support the idea that the early postnatal critical period is important.

We thank Reviewer 2 for suggesting this experiment, as we were also curious to know if fluoxetine had a similar age-specific effect on excitatory synapses based on the observed age-dependent effect of 2-photon 5-HT uncaging on sLTP in the original manuscript (now **Fig 2c, d**). To address this, we orally treated older pups with fluoxetine from P16-24, and interestingly, we no longer observed fluoxetine-mediated alterations of excitatory synapses (PMID: 36030255). Past studies indicate that 5-HT bath application has a differing effect on pyramidal neurons from slices ages P6-14 as compared to pyramidal neurons from more mature slices. Prior to P15, 5-HT depolarizes pyramidal neurons through activation of 5-HT2AR and 5-HT7Rs. After P15, 5-HT hyperpolarizes pyramidal neurons through an increased activation of 5-HT1AR signaling and decreases in 5-HT2AR and 5-HT7R mediated signaling (PMIDs: 15152041, 14742723, 17535909). We speculate that our results suggest a change in 5-HTergic receptor expression in the PFC during this slightly older developmental period, although further experiments are needed to determine this. This new data further supports the idea that there is an age-dependent effect of 5-HT on excitatory synapse development in the PFC.

We have added these new results and discussed this point in the revised manuscript (Supplementary Fig. 17; lines 284-287, 362-382).

Reviewer #3:

We thank the Reviewer 3 for their constructive comments and appreciate they found our work (1) important in the field as our findings highlight the critical role of serotonin as a mediator of PFC development, (2) well-designed, and (3) significant. We have taken into consideration all the concerns and have added many new experiments to further support our claims. The revised manuscript includes evidence using 2-photon imaging, electrophysiology, and GRAB 5-HT1.0 sensor to verify that our chemogenetic manipulations are altering the activity of 5-HTergic neurons in the dorsal raphe nucleus that innervate the prefrontal cortex (PFC) and directly affecting 5-HT release in the PFC. Furthermore, we re-analyzed morphological features from our *in vivo* DREADDs experiments to better understand the effects of 5-HTergic signaling on spine shape. Additional pharmacological experiments including alternative 5-HT2AR and 5-HT7R antagonists and 5-HT2AR and 5-HT7R selective agonists were added to verify the role of both receptors in 5-HTergic activity-mediated excitatory synaptic plasticity. Furthermore, our discussion section has been expanded significantly as a full article format to discuss current literature along with our findings.

Comments:

In this study, Ogelman et al. explored the effects of serotonin on layer 2/3 pyramidal neurons in the prefrontal cortex, during early development. First, the authors showed that chronic chemogenetic activation and inhibition of SERT-positive neurons increased and decreased spine density of layer 2/3 neurons in the PFC. These results correlate with those obtained with chronic fluoxetine administration, which is known to increase serotonin level and alter spine density. Using an elegant approach of serotonin uncaging, the authors demonstrated that serotonin induces both structural and functional LTP through the activation of 5-HT2ARs and 5HT7Rs. In contrast, new spine stabilization requires 5-HT7R activation selectively. These findings are important in the field as they highlight the critical role of serotonin as a mediator of PFC development.

Overall, the experiments are well designed, and the results are potentially significant and support most of the claims that are made. However, there are several important loose ends the authors should address, and additional control experiments should be performed. The discussion is poorly written as the authors summarized their findings, but almost did not mention current literature supporting/challenging their results. Finally, the manuscript will benefit from a schematic diagram highlighting the contribution of both receptors (i.e., localization, role, signaling).

Major comments:

1.1) Throughout the paper, the authors analyzed spine density and spine stabilization in layer 2/3 pyramidal neurons of the PFC. Spines are classified as small, medium and large relative to the fluorescence intensity. It is unclear how this criterion has been defined. A deeper analysis should be performed to evaluate the effect of serotonin on mature versus immature spines, using the standard classification of spine shape (e.g., filopodium, thin, stubby, mushroom). Moreover, the authors did not mention whether the analysis was performed on basal and/or apical dendrites and the potential differences, as these spines receive different inputs during development. All these details should be taken into consideration in order to interpret the results on both sLTP and fLTP.

We thank Reviewer 3 for their suggestion for deeper analysis of spine morphology. Using our 2-photon live imaging system, we do not have the spatial resolution to accurately classify spines based on precise shape (thin, stubby, mushroom) as 2-photon microscopy has a diffraction limited resolution of 400-500 nm and spine width and length can be less than 500 nm (PMIDs: 24847215, 28924658, 29932052). Nonetheless, to evaluate the effect of serotonin on spine structure in the

PFC, we have now analyzed spine morphology in DREADDs experiments by reporting spine length to spine head width ratio. These measurements do not provide specifics about spine shape but allow us to better determine if there are any overall changes in spine morphology (PMIDs: 37433966, 33951422, 21865455, 33109633). As spines mature, the spine length to width ratio gets smaller, and is therefore a proximate indicator of spine maturation changes (PMID: 28957675, 23269840). Spine length/width ratio was defined as the ratio of the length from the tip of the spine head to the spine neck base (spine length) to the width across the spine head at its widest point (spine width) or, if the widest point was ambiguous, at the maximum dimension perpendicular to spine length (PMID: 21865455). With the spine length/width ratio, we were also able to identify filopodia as long, thin protrusion (ratio of head width to neck width, < 1.2 and ratio of spine length to neck width > 3) (PMIDs: 34873044, 26531852). Interestingly, we observed prefrontal spines in hM4D(Gi) expressing, CNO treated mice show an immature morphology with increased filopodia, while spine morphology and filopodia percentage were unaffected in hM3D(Gq) expressing, CNO treated mice.

We have added the new analysis to Supplementary Fig. 3.

We have revised the results section to address this point (lines 97-99, 108-110).

...the authors did not mention whether the analysis was performed on basal and/or apical dendrites

All experiments were performed on apical dendrites as we mentioned in the original manuscript (i.e. *secondary and/or tertiary apical dendrites 50-100 μm from the soma*). Because layer 2/3 pyramidal neurons in the frontal cortices express 5-HT₂ARs (PMIDs: 9435262, 20802802, 24904298) and 5-HT₂ARs are densely expressed in apical dendrites of PFC pyramidal neurons (PMIDs: 11739593, 9435262), region-specific effects (apical vs basal) of 5-HT₂ergic signaling on excitatory synapse plasticity could be interesting future experiments.

We have further clarified this point in the methods and discussed possible region and layer-specific 5-HT₂ergic effect in the revised manuscript (lines 375-382, 592-595).

1.2) In Figure 1, the authors attempted to test the effect of chronic serotonin release. The authors used inhibitory and excitatory DREADDs to increase and decrease the activity of SERT-positive neurons infected in the dorsal raphe, respectively. The authors confirmed that bath application of CNO decreases and enhances the firing of hM4D(Gi)- or hM3D(Gq)-expressing DRN neurons, respectively. However, the authors never showed the expression of DREADDs in the DRN (widefield image), their colocalization with SERT, and the expression in serotonergic fibers that innervate the PFC. More importantly, the authors should test the effect of acute CNO application in the PFC (as in Supplementary 1) because their current manipulation is not specific for the serotonergic fibers that innervate the PFC. Finally, the authors used CNO to activate DREADDs, however the observed effects could arise from back-metabolism to clozapine or binding to endogenous neurotransmitter receptors, including 5-HT_{2A} and 5-HT₇ receptors. As an alternative, the authors should consider using a different DREADD agonist (e.g., DREADD agonist 21) or optogenetic stimulation of SERT-positive fibers originated from DRN (Hyun et al. 2023 PMID: 36945634).

We thank Reviewer 3 for this insightful and constructive comment. The points made by Reviewer 3 are valid concerns and have all been addressed in our revised manuscript. We appreciate all the suggestions for DREADDs verification, as they have reinforced the claims of our manuscript. Each concern has been thoroughly addressed as following:

...the authors never showed the expression of DREADDs in the DRN

Although we initially showed mCherry-tagged DREADDs expression in the DRN, Reviewer 3 is correct that we did not show a widefield image. Widefield images of retrogradely-expressing tdTomato and DREADDs-mCherry in the dorsal raphe nucleus (DRN) of SERT-Cre mice have now been added to the revised manuscript. We have also added a stack of 2-photon images of retroAAV(rAAV)-tdTomato expressing DRN 5-HT neurons as a movie to verify the efficacy of retro-viral infection in SERT-Cre mice during early development. Our images showing widefield images of rAAV-tdTomato expressing, Retro-Beads positive DRN neurons further confirm that SERT positive neurons projecting to the PFC from the DRN express Cre dependent retro-grade viruses.

We have added these new results to the revised manuscript in Supplementary Fig. 1.

Together with additional DREADD verification (*please see below*) by (1) 2-photon imaging of the fluorescent 5-HT sensor GRAB 5-HT1.0, (2) mEPSCs measurements from layer 2/3 pyramidal neurons in the PFC of hM3D(Gq)-expressing mice after CNO acute bath application, and (3) electrophysiological recordings of mCherry-positive, DREADD-expressing DRN 5-HT neurons, rAAV expression (tdTomato and mCherry) in SERT-Cre mice strongly support our claim that rAAV-DREADDs express in DRN 5-HT neurons projecting to the PFC of SERT-Cre mice.

...expression in serotonergic fibers that innervate the PFC

We have added *in vivo* and *ex vivo* 2-photon images of rAAV-tdTomato expressing 5-HTergic fibers in the PFC to further support the efficacy of retro-viral infection in SERT-Cre mice. These new images are shown in **Supplementary Fig. 1a-c**.

... authors should test the effect of acute CNO application in the PFC

Reviewer 3 raises an interesting point that acute CNO application to the PFC of DREADDs expressing mice could have a similar effect as 5-HTergic signaling on excitatory synapses. We have addressed this with the addition of multiple new experiments.

First, we verified that chemogenetic modulation of PFC-projecting raphe neurons alters 5-HT release in the PFC by using 5-HT sensor GRAB 5-HT1.0 (addgene: 140552; PMID: 33821000). To do this, we injected AAV-GRAB 5-HT1.0 into the PFC of SERT-Cre mice at P1 and acute PFC slices were made at P14-15. Using 2-photon microscopy, we confirmed that GRAB 5-HT fluorescence is increased by bath application of 5-HT (30 min, 10 μ M; PMID: 33821000). We then co-injected AAV-GRAB 5-HT1.0 and rAAV-DIO-hM3D(Gq) into the PFC of SERT-Cre at P1 and examined GRAB 5-HT fluorescence changes in acute PFC slices at P14-15 after CNO bath application (30-40 min, 1 μ M). CNO bath application significantly increased GRAB fluorescence. To ensure CNO alone does not induce off-target increases in 5-HT sensor fluorescence, we measured GRAB fluorescence changes after CNO bath application in non-hM3D(Gq) injected mice. We also measured spontaneous changes in GRAB 5-HT sensor fluorescence without CNO treatment. Neither caused any significant changes in GRAB 5-HT fluorescence. These experiments confirm that CNO bath application onto acute PFC slices of hM3D(Gq)-expressing mice induces 5-HT release (**Supplementary Fig. 6**).

We then verified hM3D(Gq) DREADDs by measuring mEPSCs from layer 2/3 pyramidal neurons in acute PFC slices of hM3D(Gq) expressing mice after CNO bath application at P11-15. CNO bath application induced a rightward shift in mEPSC amplitude. Please note that a similar increase in mEPSC amplitude was observed in 5-HT treated PFC slices (10 μ M; shown in the original manuscript, now in **Fig. 10**). Together, these data further confirm the efficacy and specificity of

DREADDs expression in SERT-Cre mice and the effect of endogenous 5-HT on prefrontal excitatory synapses.

We have added the new data to Supplementary Fig. 11.

Within the DRN itself, we confirmed DREADDs functionality via multiple experiments. We increased the range of ages for all of our DREADDs electrophysiology verification experiments from P13-15 in our original manuscript to P6-15 in the revised manuscript to mimic the ages of oral CNO administration *in vivo* (CNO from P6-15). We demonstrate that CNO bath application (30-60 mins, 1 μ M) onto hM3D(Gq) and hM4D(Gi) expressing DRN slices decreases and increases rheobase for action potential firing, respectively. Resting membrane potentials were also measured from both groups following CNO bath application. Recordings from rAAV-DREADDs-mCherry positive neurons in the DRN demonstrate that there is a functional circuit in early brain development.

We have added the new results to Supplementary Fig. 1f-j.

While we do not show colocalization with SERT, the SERT-Cre line that we use for the current study has been widely used and confirmed to be specific to SERT positive neurons in several studies (PMIDs: 17855595, 36945634, 31604921, 33186549, 15763133).

As an alternative, the authors should consider using a different DREADD agonist

Thank you to Reviewer 3 for pointing out a potential off-target effect of CNO, specifically onto 5-HT receptors. To overcome this concern, we used 2-photon microscopy to verify that CNO oral treatment *in vivo* alone does not have any effect on spine density or size in the PFC. We administered CNO from P6-15, and saline vehicle to age and gender-matched littermates. CNO alone did not alter spine density or size in comparison to saline treated controls. While this does not rule out that there may be off-target binding of CNO, these data indicate that the bidirectional changes in excitatory synapse number and strength we observed in our excitatory and inhibitory DREADDs experiments (**Fig. 1**) were not due to off-target effects of CNO.

We have added the new results to Supplementary Fig. 7.

1.3) The authors used 5-HT uncaging to test the acute effect of serotonin at the spine level. However, some clarifications and improvements are required to fully appreciate and clearly interpret the results. All data looking at spine size upon 5-HT uncaging should showed the baseline (i.e., before 5-HT HFU). Moreover, it is unclear how the authors determined the protocol of 5-HT HFU. Does it mimic physiological release of serotonin at this developmental stage?

We thank Reviewer 3 for their concern that our 5-HT HFU protocol should be described more clearly. Clarifying the exact details of our 5-HT HFU protocol is important to lay the foundation for future experiments to use this method. We have addressed the following points:

All data looking at spine size upon 5-HT uncaging should showed the baseline

Indeed, our structural plasticity (i.e. sLTP) data in the original manuscript were measured compared to the baseline. Baseline spine sizes for all experiments were calculated from the average of two baseline images (10-15 min apart) prior to 5-HT uncaging. All sLTP data are shown as % baseline after 2-photon uncaging stimulation, as previously reported (PMIDs: 36395772, 30643148, 27516412, 25558061, 23269840). We calculated our baseline spine size this way (average of two)

to account for any fluctuation in our 1 μm steps of 2-photon imaging that may underestimate or overestimate baseline spine size at one single time point.

We have clarified this point in the methods of the revised manuscript (lines 677-683).

...it is unclear how the authors determined the protocol of 5-HT HFU. Does it mimic physiological release of serotonin at this developmental stage?

We chose a 1 Hz stimulus as our 5-HT High Frequency Uncaging protocol (HFU, 30 pulses of 1 ms duration at 1 Hz) as it is within the physiological range of 5-HTergic neuronal firing (PMIDs: 22076606, 27536220). In the revised manuscript, we include a new experiment using a low frequency (0.1 Hz) 2-photon 5-HT uncaging protocol with the same number of pulses, which we are defining as Low Frequency Uncaging (LFU, 30 pulses of 1 ms duration at 0.1 Hz). 5-HT LFU did not induce structural LTP (sLTP), showing the importance of pattern specific effects of 5-HT release. The number of pulses, 30, is based on glutamatergic LTP uncaging protocols (PMIDs: 36395772, 30643148). High Frequency Stimulation (HFU) has been defined as 30-60 pulses at 1 Hz in several published works employing 2-photon uncaging (PMIDs: 15190253, 31558759, 36395772, 30643148).

We have added the new results to Fig. 2e and clarified the pattern specificity in the revised manuscript (lines 155, 166-168).

1.4) The authors provided evidence that serotonin can mediate spine stabilization through 5-HT₇R activation, but serotonin also induces sLTP and fLTP through the activation of 5-HT₂ARs and 5-HT₇Rs. However, the results rely entirely on pharmacology, and the 5-HT₂AR and 5-HT₇R antagonists may not be very selective. It is necessary that the authors use more than one drug to block 5-HT₂ARs and 5-HT₇Rs. They should also consider using knockout mice for these receptors. Moreover, it is important to know whether selective activation of 5-HT₂ARs and/or 5-HT₇Rs (using selective agonists) mimics these effects.

We thank Reviewer 3 for their insight and concerns of our pharmacological results using 5-HT₂AR and 5-HT₇R antagonists. These concerns have been thoroughly addressed by the following experiments:

... necessary that the authors use more than one drug to block 5-HT₂ARs and 5-HT₇Rs

We repeated sLTP experiments using alternative antagonists for 5-HT₂ARs (MDL11939, 1 μM) and 5-HT₇Rs (DR4485 hydrochloride, 5 μM), which are the same antagonists we used for our *in vivo* oral administration experiments. The combination of these two antagonists blocked 5-HT HFU sLTP, confirming the requirement of 5-HT₂AR and 5-HT₇R activation in 5-HTergic potentiation. We have also added experiments monitoring surface AMPAR expression using SEP-GluA2 expressing layer 2/3 pyramidal neurons in PFC slices to confirm that 5-HT₂AR and 5-HT₇R antagonists block functional LTP.

We have added the new data to Fig. 3f, g and Fig. 4g, h (and Supplementary Table 3).

... also consider using knockout mice for these receptors.

Using knockout mice would be challenging, unless PFC specific knockouts for 5-HT₂ARs and 5-HT₇Rs are used, as there could be differences in the effects of circuit level 5-HT vs post-synapse level 5-HT on excitatory synapse development. Knockout of these two critical receptors could greatly compromise the health of young pups (PMIDs: 11960784, 12529502) and could lead to

many confounding variables that would make results challenging to interpret, such as whether knockout effects are caused by circuit, cellular, and/or post-synaptic level knockout of 5-HT2A and/or 5-HT7Rs.

... whether selective activation of 5-HT2ARs and/or 5-HT7Rs (using selective agonists) mimics these effects.

This was another excellent suggestion. We recorded mEPSCs from layer 2/3 pyramidal neurons in the PFC after bath application of 5-HT2AR and 5-HT7R agonists (60-90 mins, TCB-2, 10 μ M; AS-19, 5 μ M, respectively) in the presence of TTX and CPP. Acute treatment increased mEPSC amplitude and frequency, confirming that activation of these two receptors is sufficient to induce excitatory synapse strengthening in the developing PFC.

This new data set is shown in Supplementary Fig. 15.

1.5) While activation of DRN neurons using hM3D(Gq) increases spine density in layer 2/3 pyramidal neurons of the PFC, the authors showed that uEPSC are also enhanced. Which kind of spine were targeted for uncaging (i.e., mature or immature)? uEPSC decays look different (e.g., for the traces showed in Fig. 1f and 1j).

As mentioned in our spine morphological data section (Major comment 1.1), we do not have the imaging resolution using 2-photon microscopy (due to a diffraction limited resolution of 400-500 nm) to determine specific spine shape. Nonetheless, we can estimate spine size by our 2-photon system (PMIDs: 28426965, 23269840, 37433966, 33951422, 21865455, 33109633). Spine brightness measurements provide an accurate estimate of relative synapse volume when compared with electron microscopy (PMID: 15664179) and spine size and synaptic strength are strongly correlated (PMID: 28426965). For all of our 5-HT HFU experiments, we targeted all spine sizes except for very large spines. Large spines were omitted due to our 1-photon 5-HT uncaging data showing that small and medium sized spines undergo 5-HT-dependent sLTP, but large spines do not (now **Supplementary Fig. 12a-d**). It has also been reported that small and medium sized spines are more enriched with 5-HT2A receptors (PMID: 19889983). uEPSC target spine sizes were comparable between DREADDs groups as reported in the original manuscript (now **Supplementary Fig. 4a, b**). **Supplementary Table. 2** lists target spine sizes for every experiment, confirming that 5-HT HFU target spines are statistically the same size. We did not observe any differences in the decay time (PMID: 27800545) of uEPSCs in DREADDs work [CNO only, 10.4 +/- 0.98 ms; hM4D(Gi), 10.83 +/- 1.68 ms; hM3D(Gq), 10.32 +/- 0.95 ms, n=38, 31, 32 spines, respectively].

Representative traces have been updated in Fig. 1.

1.6) In Figure 4, the authors treated fluoxetine mice chronically with 5-HT2AR and 5-HT7R antagonists. What is the effect of these antagonists administrated alone?

In our original manuscript, we reported the effect of 5-HT2AR and 5-HT7R antagonists alone (now **Fig. 6j-m**). The results from this experiment were surprising, as we did not observe any differences in spine density, size, or strength in antagonists administrated mice compared to vehicle controls. There are several interpretations for these results. First, antagonists were administered orally, thus reaching the entire nervous system and inhibiting 5-HT2ARs and 5-HT7Rs in many locations that may innervate and modulate the PFC development. Additionally, in our antagonists only experiments, drugs were only administered once per day, which may have not been sufficient to block enough 5-HT2AR and 5-HT7R-dependent 5-HTergic signals to alter spine density or strength.

Please note that in the *fluoxetine + antagonists* experiments, antagonists were administered 10 to 20 mins prior to fluoxetine delivery.

We have revised the discussion to address this point (lines 407-417).

1.7) The authors focuses on 5-HT2ARs and 5-HT7Rs. However, other serotonin receptors are present in the PFC at this development stage and may be critical for spine maturation.

We have significantly expanded our discussion section to discuss other serotonin receptors including inhibitory 5-HT receptors such as the 5-HT1A receptor that could also modulate PFC development.

We chose 5-HT2A and 5-HT7 receptors to target for our experiments for several reasons that are now clearly stated in our revised manuscript. Although several subtypes of 5-HT receptors are expressed in the PFC, we identified 5-HT2ARs and 5-HT7Rs as highly enriched in the early postnatal developing PFC and located post-synaptically (PMIDs: 9435262, 29033796, 10462127, 19889983). Past studies showed that pyramidal neuron depolarization by 5-HT in the PFC during early postnatal development is induced by a combination of 5-HT2AR and 5-HT7R activation (PMIDs: 15152041, 14742723, 17535909). Furthermore, 5-HT2ARs and 5-HT7Rs are excitatory receptors coupled to either $G\alpha_q$ or $G\alpha_s$ downstream signals that are known to upregulate synaptic plasticity (PMIDs: 22378867, 10462127, 27013076, 19889983, 36030255). Since we observed spine density increase, spine enlargement, and spine strengthening in response to increased 5-HT by both chemogenetics and 5-HT uncaging, and molecules downstream of $G\alpha_q$ or $G\alpha_s$ activation are critical for spine stabilization and growth (PMIDs: 28737723, 32105741), we chose 5-HT2AR and 5-HT7Rs to target for our pharmacological experiments.

We have clarified this point in the introduction, results, and discussion sections of the revised manuscript (lines 59-66, 176-177, 180-182, 348-375).

1.8) In the abstract, the authors mentioned the effect of serotonin “during the critical period”. However, they never defined it and they never discussed it. Are these effects different later during PFC maturation or adulthood?

We fully agree with Reviewer 3. Our study focuses on the maturation and stabilization of dendritic spines in the context of circuit formation during early postnatal development. Many studies have shown that the first two weeks of postnatal development in mice are essential for laying the foundation for circuit development, hence defined as a critical period for synapse development in many published works (PMIDs: 8895456, 16261181, 2573152, 7753197, 33951422). In our revised manuscript, we defined and discussed the developmental stage-specific effect of 5-HT on excitatory synapses with additional data. We observed an age-dependent effect of 5-HT uncaging on structural LTP in the original manuscript (now in **Fig. 2c, d**), and during revision we examined if fluoxetine has a similar age-specific effect on excitatory synapses. To do this, we orally treated older pups with fluoxetine from P16-23, and interestingly, we no longer observed fluoxetine-mediated alterations of excitatory synapses (PMID: 36030255). Past studies indicate that 5-HT bath application has a differing effect on pyramidal neurons from slices ages P6-14 as compared to pyramidal neurons from slightly more mature slices. Prior to P15, 5-HT depolarizes pyramidal neurons through activation of 5-HT2AR and 5-HT7Rs. After P15, 5-HT hyperpolarizes pyramidal neurons through an increased activation of 5-HT1AR signaling and decreases in 5-HT2AR and 5-HT7R-mediated signaling (PMIDs: 15152041, 14742723, 17535909). We speculate that our results suggest a change in 5-HTergic receptor expression in the PFC during this slightly older developmental period, although further experiments are needed to determine this. This new data further supports the specific effect of serotonin on PFC development during the critical period.

We have added these new results (Supplementary Fig. 17) and discussed this point in the revised manuscript (lines 284-287, 362-382).

1.9) The discussion is poorly written as the authors summarized their findings, but almost did not mention current literature supporting/challenging their results. Finally, the manuscript will benefit from a schematic diagram highlighting the contribution of both receptors (i.e., localization, role, signaling).

We agree with Reviewer 3. Because our original manuscript was automatically transferred from another journal, which requires a brief format, to the current *Nature Communications*, our discussion indeed needs more details on 5-HTergic receptor signaling during development.

We have revised the entire discussion section.

We have revised our original schematic diagram (now in **Fig. 5r**). More importantly, we have also expanded our discussion significantly to address 5-HT7R and 5-HT2AR dependent pathways and their involvement in spine maintenance and potentiation. 5-HT7Rs signal through $G\alpha_s$ coupled proteins, which activate cAMP-dependent voltage gated Ca^{2+} channels (VGCCs) and increase extracellular Ca^{2+} entry into spines. In newly formed spines, the increase in Ca^{2+} through 5-HT7R dependent mechanisms may be sufficient to increase stabilization (PMIDs: 25498985, 18667153, 21295598, 23303946, 22334212). 5-HT2ARs signal through $G\alpha_q$ coupled proteins, notably activating PKC downstream. We believe that during 5-HT induced LTP, increased Ca^{2+} entry through 5-HT7R VGCC activation contributes to 5-HT2AR-dependent PKC activation, as PKC requires Ca^{2+} to be activated (PMIDs: 30013171, 30013172, 2479143) during synaptic potentiation.

We have revised the discussion to emphasize this point (lines 335-361).

Minor comments:

1.10) For experiments measuring uEPSCs, the authors should show the individual spines, not only the mean.

In our revised manuscript, we have added all data points for uEPSC amplitudes.

1.11) The authors should report whether chronic manipulations, such as DREADDs and fluoxetine, influence dendritic arborization and complexification (Sholl analysis) as well as intrinsic properties of layer 2/3 pyramidal neurons.

We thank Reviewer 3 for this suggestion. Our revised manuscript further shows firing properties and resting membrane potential (RMP) of layer 2/3 pyramidal neurons in the PFC after DREADDs manipulations. We found no changes in RMP or excitability in either hM3D(Gq) or hM3D(Gi) group. We did not measure intrinsic properties after fluoxetine manipulations because oral fluoxetine treatment can affect the entire nervous system including many different brain circuits that innervate the PFC, making results challenging to interpret. Our DREADDs manipulations are specific to 5-HTergic projections into the PFC, and we therefore believe these new results suggest that alterations to this pathway in our experimental system do not affect intrinsic properties of prefrontal layer 2/3 pyramidal neurons. This further suggests that 5-HTergic activity in the PFC regulates synaptic plasticity in a glutamatergic activity-independent manner.

This new data set is shown in Supplementary Fig. 9.

It will be very interesting to investigate on the dendritic level, how serotonin modulates dendritic arborization based on past work on the role of 5-HT in dendritic growth (PMIDs: 28974382,

24752854, 27894797, 31292861) through the activation of small GTPases (PMID: 10700252, 26139370). Our future work will focus on unraveling the dendritic mechanisms in experiments outside the scope of the current manuscript that is focused on the synaptic mechanisms.

1.12) The authors did not mention whether they use males and/or females.

We thank Reviewer 3 for the concern that sex selection was not stated clearly enough in the original manuscript that only mentioned “*Acute coronal slices of the PFC were prepared from both male and female wild type and SERT-Cre mice...*”. We have added more information to our methods section to clearly state that both males and females were used. Additionally, in the revised work, we measured resting membrane potentials of hM3D(Gq) or hM4D(Gi) expressing 5-HTergic neurons in the raphe nuclei for males and females after CNO treatment and found no sex-differences between the two groups (**Supplementary Fig. 1h and j**). We also found a comparable increase in the amplitude of mEPSCs induced by CNO-mediated 5-HT release in the PFC of male and female mice. For PFC slice culture work, we made slice cultures using both male and female pups at P3-4 and slices were randomly used for experiments at EP11-14 [equivalent postnatal (EP) day = postnatal day at slice culturing + days *in vitro*]. We thus cannot conclude whether there are sex differences in slice culture data. Future studies looking further into whether there are any age-dependent sex effects will be critical next steps in this work.

We have revised the results and discussion sections to address gender differences in our work (lines 135-140, 144-145, 382-389, 554, 572, 729).

1.13) The authors did not mention which part of the PFC they focused on.

Thank you to the Reviewer 3 for bringing this point to our attention. The region of the PFC that was injected with AAVs (dorsolateral PFC) was mentioned in our original manuscript, but it was not clear that our imaging and recording data were collected from the same region. In our revised manuscript we more clearly state that all data collected from PFC acute slices and PFC slice cultures was from layer 2/3 pyramidal neurons in the dorsolateral PFC. We added an interesting new data set demonstrating that 5-HT HFU onto excitatory synapses of layer 5 pyramidal neurons does not induce sLTP, suggesting a possible region and layer-specific 5-HTergic effect.

We have revised the results and method sections to clarify the brain region (lines 82, 105, 583, 630, 710) and discussed possible region and layer-specific 5-HTergic effect in the revised manuscript (lines 375-382).

1.14) The authors may want to revise the title as the study focuses on the effect of serotonin on dendritic spine during PFC development.

We thank the Reviewer for suggesting a revised title for our manuscript. With more data including functional LTP and age-specificity, our new title is *Serotonin modulates excitatory synapse maturation in the developing prefrontal cortex*.

REVIEWERS' COMMENTS

Reviewer #1 (Remarks to the Author):

The authors have adequately addressed previous concerns.

Although I am still a bit confused about the relationship between sLTP, fLTP, and spine survival, in which the authors' new results show SEP-GluA2 is only inserted in survived new spines, and blocking 2AR and 7R also block fLTP, but blocking 2AR does not affect new spine survival. However, these discrepancies could be due to the specificity or the global application of the drugs as the authors mentioned in the discussion.

Reviewer #2 (Remarks to the Author):

The authors fully addressed my review comments. I do not have additional comments.

Reviewer #3 (Remarks to the Author):

The authors have properly address my concerns by adding new experiments and editing the text. This elegant study is important in the field.

I only have a suggestion regarding Fig. 2. Perhaps the title of Fig. 2 should mention the difference between layer II/III and layer V pyramidal neurons, as this is a very interesting finding.

NCOMMS-23-17520A

Reviewer Comments:

Reviewer #1

We thank Reviewer 1 for their thoughtful comment.

Comments:

The authors have adequately addressed previous concerns.

Although I am still a bit confused about the relationship between sLTP, fLTP, and spine survival, in which the authors' new results show SEP-GluA2 is only inserted in survived new spines, and blocking 2AR and 7R also block fLTP, but blocking 2AR does not affect new spine survival. However, these discrepancies could be due to the specificity or the global application of the drugs as the authors mentioned in the discussion.

Reviewer #2

We thank Reviewer 2 for their positive remarks.

Comments:

The authors fully addressed my review comments. I do not have additional comments.

Reviewer #3

We appreciate Reviewer 3 for their constructive comment. The title of Fig.2 is revised accordingly.

Comments:

The authors have properly address my concerns by adding new experiments and editing the text. This elegant study is important in the field.

I only have a suggestion regarding Fig. 2. Perhaps the title of Fig. 2 should mention the difference between layer II/III and layer V pyramidal neurons, as this is a very interesting finding.